# LauraGPT: Listen, Attend, Understand, and Re-generate Audio with GPT

## Abstract

Generative Pre-trained Transformer (GPT) models have achieved remarkable performance on various natural language processing tasks. However, there has been limited research on applying similar frameworks to audio tasks. Previously proposed large language models for audio tasks either lack sufficient quantitative evaluations, or are limited to tasks for recognizing and understanding audio content, or significantly underperform existing state-of-the-art (SOTA) models. In this paper, we propose **LauraGPT**, a unified GPT model for audio recognition, understanding, and generation. LauraGPT is a versatile language model that can process both audio and text inputs and generate outputs in either modalities. It can perform a wide range of tasks related to content, semantics, paralinguistics, and audio-signal analysis. Some of its noteworthy tasks include automatic speech recognition, speech-to-text translation, text-to-speech synthesis, machine translation, speech enhancement, automated audio captioning, speech emotion recognition, and spoken language understanding. To achieve this goal, we use a combination of continuous and discrete features for audio. We encode input audio into continuous representations using an audio encoder and decode output audio from discrete codec codes. We then fine-tune a large decoder-only Transformer-based language model on multiple audio-to-text, text-to-audio, audio-to-audio, and text-to-text tasks using a supervised multitask learning approach. Extensive experiments show that LauraGPT achieves competitive or superior performance compared to existing SOTA models on various audio processing benchmarks.

## 1 Introduction

One of the main goals of artificial intelligence (AI) research is to achieve artificial general intelligence (AGI), which is the hypothetical ability of a machine to perform any intellectual task that a human can do. A popular approach to pursue this goal is to use large language models (LLMs), which are neural networks that generate natural language texts based on a given context. LLMs can learn from massive amounts of text data and mimic human language to acquire most human knowledge. LLMs such as GPT-4 (OpenAI, 2023), PaLM2 (Anil et al., 2023), LLaMA (Touvron et al., 2023) have demonstrated impressive capabilities across various domains, exhibiting zero-shot generalization without the need for task-specific fine-tuning. However, these models are primarily limited to processing text data.

Text and audio are two important modalities for human communications. Recent research aims to create a seamless integration of text and audio through a unified audio-text model that can handle various tasks within and across these modalities. However, existing approaches have limitations in terms of **model architecture**, **data representation**, and **task coverage**.

Existing work can be categorized into four groups: (1) self-supervised learning of a universal audio encoder, (2) encoder-decoder models with modal-specific pre-nets and post-nets, (3) models converting audio features to text, and (4) decoder-only models with discrete audio tokens. We will first introduce the last two groups which are most related to our work in this section, and will dive into more details of all four categories in Section 2.

The third group of models converting audio features to text includes models that can perform multilingual automatic speech recognition (ASR) and speech-to-text translation (S2TT) such as Whisper (Radford et al., 2022) and USM (Zhang et al., 2023b), and models that can perform more kinds of audio understanding tasks, such as Pengi (Deshmukh et al., 2023). However, these

models cannot generate speech outputs from text tokens, which restricts their applications for speech generation tasks such as text-to-speech synthesis (TTS). The fourth group of models adopts a decoder-only framework after converting continuous audio into discrete tokens and merging text and audio tokens into a shared vocabulary, such as VioLA (Wang et al., 2023b) and AudioPaLM (Rubenstein et al., 2023). These models can perform ASR, TTS, S2TT, machine translation (MT), and speech-to-speech translation (S2ST) tasks with a single model, enabling a more natural and flexible interaction between text and audio. They can generate speech outputs from text or speech inputs by using codec-based discrete features (Zeghidour et al., 2022; Défossez et al., 2022). However, they may suffer from the information loss caused by quantization of speech signals into discrete tokens, which leads to significant performance degradation over models using continuous speech features, as demonstrated in our ablation study that using continuous features for audio input greatly outperforms using discrete features on ASR, S2TT, and speech enhancement (SE) tasks (Section 5.2).

In this paper, we propose a novel unified audio-text LLM that overcomes these limitations of the existing approaches. Our model adopts a decoder-only Transformer framework, which offers *enhanced simplicity and generality* compared to the Transformer encoder-encoder framework. We introduce a unique data representation that combines the strengths of both continuous and discrete representations of audio signals. Specifically, we use continuous features to represent the input audio, which ensures performance of audio comprehension tasks, and use codec-based discrete features to represent the output audio, which enables joint autoregressive modeling with text features for audio generation tasks. Experimental results reveal that continuous features for audio (such as Filterbank) have notable advantages over discrete units on audio recognition, understanding, and audio-signal related tasks, such as ASR, S2TT, and SE. Overall, our model achieves a better trade-off over existing approaches between model uniformity and high performance on diverse categories of audio tasks.

Our paper makes three main contributions [1]:

- We propose **LauraGPT**, a unified audio-text LLM under the GPT framework, which uses a novel data representation that **combines continuous and discrete features for audio signals**. This data representation preserves both fidelity and generality of audio data and enables joint modeling with text features.
- We conduct multi-task fine-tuning of our unified model on diverse audio tasks. Specifically, we reformulate speech-signal-related tasks such as speech enhancement and paralinguistics-centered tasks such as speech emotion recognition (SER) as sequence-to-sequence modeling. Furthermore, we extend our model to support general audio tasks such as automated audio captioning (AAC) and semantics-oriented tasks such as spoken language understanding (SLU). To the best of our knowledge, our single model supports **the largest number of and most diverse audio processing tasks** among existing unified audio-text models that focus on **audio recognition, understanding, and generation**.
- We conduct extensive experiments on multiple benchmarks and show that **our proposed LauraGPT trained with open source data achieves competitive or superior performance compared to SOTA models on all tasks**. Our model effectively strikes a balance between model uniformity and high performance on diverse audio processing tasks.

## 2 RELATED WORK

### 2.1 UNIFIED AUDIO-TEXT MODELING

Existing works on unified audio-text modeling can be categorized into four groups: (1) self-supervised learning of a universal audio encoder, (2) encoder-decoder models with modal-specific pre-nets and post-nets, (3) models converting audio features to text, and (4) decoder-only models with discrete audio tokens. The first group of works, such as wav2vec 2.0 (Baevski et al., 2020), HuBERT (Hsu et al., 2021), and WavLM (Chen et al., 2022), can leverage unlabeled speech data for pre-training, but they require additional *task-specific models* for downstream audio tasks. The second group, such as SpeechT5 (Ao et al., 2022) and SpeechNet (Chen et al., 2021b), can perform various speech tasks with a single model, but they need to employ *modal-specific pre-nets and post-nets* to handle different input/output modalities. The third group, such as Whisper (Radford et al., 2022), USM (Zhang et al., 2023b), and Pengi (Deshmukh et al., 2023), focus on converting continuous speech features into

---

[1]Demos are available at https://lauragpt.github.io

text, but they *cannot support audio generation tasks*. The fourth group, such as VioLA (Wang et al., 2023b), AudioPaLM (Rubenstein et al., 2023), SpeechGPT (Zhang et al., 2023a) and SpeechGen (Wu et al., 2023), use decoder-only Transformers to model discrete audio tokens and text tokens as a shared vocabulary, but they may suffer from *the information loss* caused by the quantization of audio signals into discrete tokens. Table 1 compares our LauraGPT against the most related works, which, similar to LauraGPT, are all multi-task unified audio-text models. Distinct from unified audio-text modeling, some studies integrate expert audio models with LLMs to enable direct audio interaction with LLMs, such as HuggingGPT (Shen et al., 2023) and AudioGPT (Huang et al., 2023b); however these models have *high complexity*, *resource consumption*, and *error accumulation*.

Table 1: Comparisons with the most related multi-task unified audio-text models. The table shows the tasks that each model is trained and evaluated on.

|  | SpeechT5 | Whisper | VioLA | AudioPaLM | LauraGPT(Ours) |
|---|---|---|---|---|---|
| **Date** | 2021.10 | 2022.12 | 2023.5 | 2023.6 | 2023.9 |
| **Organization** | Microsoft | OpenAI | Microsoft | Google | Ours |
| **Model Size** | 0.14B | 1.5B | 0.25B | 8B | 2.0B |
| **Pair Data (hrs)** | 0.96K | 680K | 79K | 48K | 60K |
| **Unsup. Pretrain** | N/A | N/A | N/A | PaLM-2 | Qwen-2B |
| **Audio Input** | Continuous | Continuous | Discrete | Discrete | Continuous |
| **Audio Output** | N/A | N/A | Discrete | Discrete | Discrete |
| **Languages** | EN | 99 | EN/CN | 113 | EN/CN |
| **ASR** | ✓ | ✓ | ✓ | ✓ | ✓ |
| **S2TT** | ✓ | ✓ | ✓ | ✓ | ✓ |
| **TTS** | ✓ | ✗ | ✓ | ✓ | ✓ |
| **MT** | ✗ | ✗ | ✓ | ✓ | ✓ |
| **SE** | ✓ | ✗ | ✗ | ✗ | ✓ |
| **AAC** | ✗ | ✗ | ✗ | ✗ | ✓ |
| **SER** | ✗ | ✗ | ✗ | ✗ | ✓ |
| **SLU** | ✗ | ✗ | ✗ | ✗ | ✓ |

## 2.2 AUDIO GENERATION FROM TEXT PROMPTS

Two prominent categories of approaches have emerged recently for generating audio signals from text prompts. **(1)** In the first category, continuous representations such as utterance-level embeddings (Elizalde et al., 2022; Liu et al., 2023; Huang et al., 2023a) and Mel-frequency spectrograms (Nachmani et al., 2023) are used as the targets. However, continuous representations present a challenge for unified modeling of text and audio within a single language model. **(2)** In the second category, discrete codec tokens are employed as audio representations and generated by diffusion models (Yang et al., 2023) or autoregressive language models (Kreuk et al., 2023; Borsos et al., 2023; Copet et al., 2023; Wang et al., 2023a). Among them, in models such as AudioGen (Kreuk et al., 2023), AudioLM (Borsos et al., 2023), and MusicGen (Copet et al., 2023), multiple output heads are used after the language model to predict synchronized or delayed groups of codec tokens. However, this approach is only suitable for audio generation and may not be applicable to diverse audio-text tasks. In VALL-E (Wang et al., 2023a), the language model predicts output tokens of the first quantizer, while tokens of the remaining quantizers are predicted by a non-autoregressive model one by one. This mechanism requires numerous prediction procedures to achieve an acceptable quality. Moreover, the indices of the remaining codec groups are challenging to predict due to the multi-modal distribution nature of codec tokens (Jenrungrot et al., 2023). To overcome these challenges, we propose a one-step codec vocoder in our LauraGPT, where a transformer-based predictor is trained to estimate the summation of all codec token groups instead of the multi-group indices, by minimizing the reconstruction losses. Our approach reduces the audio generation process to a single feed-forward calculation and overcomes the prediction challenge arising from the multi-modal distribution nature.

## 3 METHODOLOGY

Figure 1 illustrates an overview of the proposed LauraGPT. LauraGPT comprises three components: a GPT backbone, an audio encoder, and a codec vocoder. For audio inputs, we extract the log-compressed Mel spectrogram features and feed them into the audio encoder, while the audio outputs are discretized into tokens using the audio tokenizer. As for text data, both inputs and outputs are

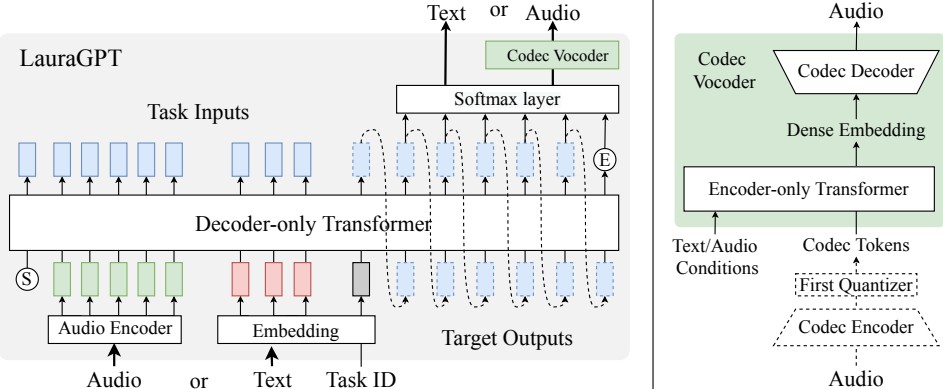

Figure 1: An overview of the proposed LauraGPT model, where Ⓢ and Ⓔ denote the "start of sequence" and "end of sequence" tokens, respectively. The right part provides an enlarged view of the Codec Vocoder in LauraGPT. Note that the dashed modules are only used at the training stage. We omit the text tokenizer and detokenizer for simplicity.

processed by the text tokenizer. We describe the pre-trained GPT backbone and text tokenizer in Section 3.1 and the audio tokenizer in Section 3.2. Section 3.3 elaborates our modifications made to the GPT backbone to enable unified audio-text modeling. Section 3.4 introduces an efficient and high-quality one-step codec vocoder for converting audio tokens into raw waveforms. Section 3.5 describes our multi-task fine-tuning approach and the paradigm for supporting more complex tasks.

## 3.1 PRE-TRAINED GPT BACKBONE AND TEXT TOKENIZER

We utilize the open-source language model, Qwen (Bai et al., 2023), as the backbone, which is pre-trained on a diverse corpus of 3 trillion tokens covering various domains in English and Chinese. Qwen models are decoder-only transformers equipped with rotary position embedding (RoPE) (Su et al., 2021). Compared with other open-source GPT models of similar model sizes, Qwen models demonstrate strong competitiveness, achieving better performance on widely used benchmarks, especially for Chinese tasks (Bai et al., 2023). To leverage pre-trained Qwen checkpoints, we employ the original Qwen tokenizer, which inherits the fast BPE tokenizer tiktoken from GPT-4 (Jain, 2022) and incorporates additional augmentations for commonly used characters and words in different languages. The text tokenizer has a vocabulary size of approximately 152K.

## 3.2 AUDIO TOKENIZER

Different from text, audio signal is commonly represented as a sequence of continuous features. This poses a challenge when integrating text and audio generation into a single GPT model. To address this challenge, we utilize a codec model as the audio tokenizer to extract discrete representations.

Our codec model shares a similar architecture as Encodec (Défossez et al., 2022), which consists of a convolutional recurrent encoder/decoder network (Tagliasacchi et al., 2020) and a residual vector quantizer (RVQ) (Vasuki & Vanathi, 2006). We enhance the original Encodec model with the following modifications. Firstly, we add reconstruction losses in the magnitude spectrum domain to **improve the quality of middle-frequency and high-frequency audio**. Secondly, to **address the challenge of long sequence lengths**, we stack five strided convolution blocks with strides of [8, 5, 4, 2, 2], resulting in a token rate of 25 Hz for each group. Thirdly, the RVQ module comprises 32 quantizers with structured dropout, each with a vocabulary size of 1024. This modification **improves speech quality** with more quantizers, **while preserving the most information in the shallow quantizers**. The encoder and the first quantizer in RVQ are used as an audio tokenizer, and the outputs of the first quantizer are treated as the audio tokens. Note that the remaining quantizers and the decoder are only used during the training stage. Other details are similar to Du et al. (2023).

### 3.3 MODIFIED LANGUAGE MODEL FOR UNIFYING AUDIO-TEXT MODELING

In text-only GPTs, the tokenized text is first passed through an embedding matrix that converts discrete tokens into dense embeddings. In other speech-text Transformers (Wang et al., 2023b; Rubenstein et al., 2023), the embedding matrix is augmented with speech tokens to unify speech-text modeling. However, our preliminary experiments revealed that using a single tokenizer to represent diverse audio signals significantly impairs the performance of all audio tasks (see Section 5.2 for more details). Therefore, to strike a balance between uniformity and performance, we use a Conformer-based encoder to convert audio inputs into *continuous* representations, while the tokenized text inputs continue to undergo embedding matrix transformation to generate dense vectors. The audio representations and text embeddings have the same dimension $D$. The Conformer-based encoder is initialized with weights from a pre-trained ASR model (Gao et al., 2023). Since batch normalization can lead to endless loop decoding, we replace it with layer normalization in the Conformer-based encoder (see Appendix B.1 for more details).

To achieve the goal of unified generation of audio and text, we augment the embedding matrix with codec tokens. The embedding matrix $\mathbf{E}$ in the final softmax layer is of size $(N + M + L) \times D$ and is utilized to calculate the logits for audio and text tokens at each position. Within $\mathbf{E}$, $N$ embeddings represent text tokens obtained from the Qwen tokenizer, $M$ embeddings are assigned to audio tokens extracted by codec models, and $L$ embeddings are allocated for special tokens (e.g., task IDs) to differentiate between different tasks. To leverage the pre-trained weights, $N$ text-related embeddings are initialized with the Qwen-2B model checkpoint. In contrast, $M$ audio-related embeddings and $L$ special token embeddings are randomly initialized.

Based on the aforementioned representations, the Qwen backbone is trained to model various audio/text tasks by minimizing the cross-entropy loss:

$$\mathcal{L}_{LM} = -\frac{1}{T_v} \sum_{j=1}^{T_v} \log p_\theta \left(\mathbf{v}_j | \mathbf{u}_1, \ldots, \mathbf{u}_{T_u}, \mathbf{u}_{task}, \mathbf{v}_1, \ldots, \mathbf{v}_{j-1}\right) \tag{1}$$

where $\mathbf{u}$ denotes the input embeddings with a sequence length of $T_u$. $\mathbf{v}$ represents the sequence of target tokens with a length of $T_v$. To specify a task, a special task-related token $\mathbf{u}_{task}$ is inserted between the input embeddings and output tokens. Note that only the losses of outputs are taken into account, while losses on inputs and task embeddings are masked out. After the final output layer, text tokens are converted to the final outputs using the Qwen tokenizer, and audio tokens are decoded to raw waveforms using a codec vocoder, as described in Section 3.4. All model parameters are jointly trained, except for the codec vocoder, which is trained independently and kept frozen during both training and inference stages of LauraGPT.

### 3.4 ONE-STEP CODEC VOCODER FOR AUDIO GENERATION

In LauraGPT, we propose a one-step codec vocoder to generate raw audio signals from the codec tokens, which are extracted with **only the first quantizer** as described in Section 3.2. The remaining quantizers are only used for training. Our vocoder comprises two components: a transformer-based predictor and a codec decoder. Alongside the predicted codec tokens from the Qwen backbone, text and audio conditions are also concatenated. For instance, the text content serves as a condition for the TTS task, while the noisy speech features are employed as conditions for the SE task. The predictor is trained to estimate the summation of token embeddings in all groups by minimizing the L1 and L2 distances between the predicted embeddings $\hat{\mathbf{E}}$ and their corresponding ground truth $\mathbf{E}$:

$$\mathcal{L}_{pre} = \frac{1}{T} \frac{1}{D_c} \sum_{t=1}^{T} \sum_{i=1}^{D_c} ||\mathbf{E}_{t,i} - \hat{\mathbf{E}}_{t,i}||_1 + 0.5 \cdot ||\mathbf{E}_{t,i} - \hat{\mathbf{E}}_{t,i}||_2 \tag{2}$$

where $T$ denotes the total number of frames, and $D_c$ denotes the dimension of the codec embeddings. After obtaining the estimated embeddings, the decoder of an pre-trained codec model is utilized to reconstruct the raw audio signals.

### 3.5 MULTI-TASK FINETUNING

**Basic Tasks**   In this work, we unify modeling of the following tasks under the GPT framework and use them for multi-task fine-tuning of LauraGPT: Automatic Speech Recognition (**ASR**), Spoken

Language Understanding (**SLU**), Speech to Text Translation (**S2TT**), Speech Emotion Recognition (**SER**), Automated Audio Captioning (**AAC**), Speech Enhancement (**SE**), Text-to-speech Synthesis (**TTS**). Task definitions can be found in Appendix A.1.

**Unified Task Expression**    LauraGPT operates based on a unified task expression: **input embeddings, task ID, output tokens**. With the same inputs, the desired outputs can differ across tasks. Task-related tokens are included in both the input and output embedding matrices. It is worth noting that in addition to masking out the losses on inputs, the cross-entropy loss at the position of the task token is also masked out. The TTS and SLU tasks take text embeddings as inputs, while the ASR, S2TT, SE, ACC, and SER tasks take audio encodings as inputs. The TTS and SE tasks use the outputs of the first quantizer as the target outputs, while the remaining tasks use text tokens.

**Support More Complex Tasks**    With its modular and flexible design, LauraGPT provides an extensible framework to support complex tasks. By breaking a task into sub-tasks among the aforementioned basic tasks and cascading the raw inputs and model outputs of sub-tasks, LauraGPT can perform more complex tasks than the basic tasks. As an example, this paper demonstrates that LauraGPT is capable of performing the advanced task of speech-to-speech translation (S2ST) by combining the S2TT and TTS tasks. Initially, a sequence is constructed to translate the speech content into another language using the S2TT task token: `[audio embedding, <S2TT>]`. Subsequently, the translated text is combined with the TTS task token to synthesize speech: `[text embedding, <TTS>]`. If maintaining the speaker identity is desired, the original inputs and content can be incorporated to perform personalized TTS. This can be achieved with an input sequence as `[ASR recognized text embedding, S2TT translated text embedding, <TTS>, codec token of raw speech]`, where `ASR recognized text embedding` is obtained using the ASR task: `[audio embedding, <ASR>]`. This approach treats the bilingual text as the complete input and allows the model to generate an output sequence of codec tokens while maintaining the same speaker identity. We provide synthesized audio samples to demonstrate this S2ST capacity at `https://lauragpt.github.io`.

## 4    EXPERIMENTAL SETTINGS

**Model Architecture** The Conformer-based encoder for extracting continuous audio representations consists of 32 conformer blocks. Each block consists of a feed-forward module with 1536 units, an attention module with 16 attention heads and an attention dimension of 512, a convolutional module including the pointwise and depthwise convolution layers, and a second feed-forward module with 1536 units. Sinusoidal positional encoding is applied on the inputs of Conformer. As explained in Section 3.3, batch normalization is replaced with layer normalization in the convolutional module. For a trade-off between performance and training efficiency, we use Qwen-2B[2] with 1.8B trainable parameters as the backbone. Qwen-2B comprises 24 transformer layers with a hidden size of 2048 and 16 attention heads. Note that although Conformer and Qwen-2B are selected as the audio encoder and GPT backbone in this work, they can be replaced by other encoders and GPT models.

**Training Setup** In all experiments, we initialize the Qwen backbone and audio encoder with the pre-trained checkpoints. We then optimize the model parameters through multi-task fine-tuning. Details of the training setup can be found in Appendix A.4. Details of the training/evaluation datasets and evaluation metrics are presented in Appendix A.2 and  A.3.

## 5    EXPERIMENTAL RESULTS AND ANALYSIS

Section 5.1 presents the main experimental results of each task. Additional results for MT are in Appendix A.6. Section 5.2 shows the advantages of using continuous features as inputs and discrete tokens as outputs in LauraGPT by comparing LauraGPT with a model trained with discrete inputs and outputs (denoted **Discrete IO**). Further analyses are in Appendix B, including comparing batch normalization with layer normalization in the audio encoder (Appendix B.1), impact of initialization from pre-trained models (Appendix B.2), and impact of multi-task fine-tuning (Appendix B.3).

### 5.1    RESULTS ON EACH TASK

**ASR Evaluation**    We evaluate the ASR performance on Chinese AISHELL and English Libirispeech test sets, as shown in Table 2. The baselines include Paraformer (Gao et al., 2022) and Whisper Large V2 (Radford et al., 2023). The recently proposed Paraformer is a competitive non-autoregressive

---

[2]https://github.com/QwenLM/Qwen

ASR model. Two open-source models Paraformer(CN)[3] and Paraformer(EN)[4] are directly used for evaluation, which are pre-trained on large-scale industrial Chinese (60K hours) and English (20K hours) datasets, respectively. We also compared with the off-the-shelf Whisper model, a multilingual ASR model trained on diverse audio datasets of 680K hours. Table 2 shows that both Paraformer and Whisper are highly competitive baselines on Chinese and English ASR tasks. Notably, LauraGPT greatly outperforms Whisper on Chinese sets by **-3.9** and **-2.3** absolute on CER and performs comparably to Paraformer(CN) with much smaller amount of training data. On the English test sets, LauraGPT achieves comparable WER to Paraformer(EN) on test-clean and test-other. However, Whisper outperforms LauraGPT, which may be primarily attributed to the much smaller English data used for training LauraGPT.

Table 2: Comparison of different models on the ASR task in terms of CER(%) ↓ for Chinese and WER(%) ↓ for English. Data size denotes the number of hours.

| Model | Model Size | Data Size | AISHELL-1 test | AISHELL-2 test-ios | Librispeech test-clean | Librispeech test-other |
|---|---|---|---|---|---|---|
| **Paraformer (CN)** | 0.2 B | 60K | 2.0 | 2.9 | - | - |
| **Paraformer (EN)** | 0.2 B | 20K | - | - | 3.5 | 8.2 |
| **Whisper Large V2** | 1.5 B | 680K | 5.7 | 5.5 | 2.5 | 4.9 |
| **Discrete IO** | 1.8 B | 22K | 7.1 | 8.6 | 9.1 | 24.0 |
| **LauraGPT (Ours)** | 2.0 B | 22K | 1.8 | 3.2 | 4.4 | 7.7 |

**SLU Evaluation** We evaluate the SLU performance of different models on the SLURP test set (Bastianelli et al., 2020), as shown in Table 3. The first group of Table 3 reports the results of two competitive baselines, CRDNN and Wav2Vec 2.0. The Wav2Vec 2.0 includes self-supervised pre-training, while the CRDNN does not. We find that LauraGPT achieves the highest accuracy among all models, indicating its superiority to understand the user's intent. It also achieves comparable results with wav2vec 2.0 on word-F1, char-F1, and SLU-F1, demonstrating its effectiveness on slot filling.

Table 3: Comparison of different models on the SLU task.

| Model | Scenario (ACC ↑) | Action (ACC ↑) | Intent (ACC ↑) | Word-F1 ↑ | Char-F1 ↑ | SLU-F1 ↑ |
|---|---|---|---|---|---|---|
| **Ravanelli et al. (2021) CRDNN** | 82.15 | 77.79 | 75.64 | 62.35 | 66.45 | 64.34 |
| **Ravanelli et al. (2021) Wav2Vec 2.0** | 89.49 | 86.40 | 85.34 | 72.60 | 76.76 | 74.62 |
| **LauraGPT (Ours)** | 91.04 | 89.07 | 87.87 | 71.20 | 75.86 | 73.45 |

**S2TT Evaluation** We evaluate LauraGPT on BSTC dev set (Zh→En) (Zhang et al., 2021) and CoVOST2 test set (En→Zh)(Wang et al., 2020), as shown in Table 4. We report the baseline results on BSTC and CoVOST2 for comparison. The baseline of BSTC is a cascaded system that includes an SMLTA ASR model and an MT model pre-trained on the WMT19 corpus (Zhang et al., 2021). The CoVOST2 baseline is an end-to-end S2TT model pre-trained on ASR (Wang et al., 2020). On the CoVOST2 test set (En→Zh), LauraGPT improves the baseline BLEU score by **+13.1** absolute. When compared to the BSTC baseline (Zh→En), LauraGPT only yields a minor -0.4 BLEU reduction. These results demonstrate the competitive performance of LauraGPT on the S2TT task.

Table 4: Comparison of different models on the S2TT task in terms of BLEU (↑).

| Model | Zh→En | En→Zh |
|---|---|---|
| **Zhang et al. (2021)** | 18.2 | - |
| **Wang et al. (2020)** | - | 25.4 |
| **Discrete IO** | 5.1 | 5.0 |
| **LauraGPT (Ours)** | 17.8 | 38.5 |

---

[3]https://www.modelscope.cn/models/damo/speech_paraformer-large_asr_nat-zh-cn-16k-common-vocab8404-pytorch/summary

[4]https://www.modelscope.cn/models/damo/speech_paraformer_asr_en-16k-vocab4199-pytorch/summary

**SER Evaluation** We evaluate the SER performance on the MELD test set (Poria et al., 2018), as shown in Table 5. Considering the imbalanced data distribution of the MELD dataset, we report the weighted F1 score (WF1) as the primary metric, together with weighted accuracy (WA) and unweighted accuracy (UA). We take the WavLM model (Chen et al., 2022) and the recently developed SER-specialized Vesper-L2 model (Chen et al., 2023) as the competitive baselines. Table 5 shows that LauraGPT improves WF1 and UA over the baselines, demonstrating its superiority on SER task. Further details regarding SER evaluation can be found in Appendix A.5.

Table 5: Comparison of different models on the SER task. WF1 is the primary metric.

| Model | WA ↑ | UA ↑ | WF1 ↑ |
|---|---|---|---|
| **Chen et al. (2023) WavLM Base** | 0.499 | 0.201 | 0.400 |
| **Chen et al. (2023) WavLM Large** | 0.542 | 0.253 | 0.476 |
| **Chen et al. (2023) Vesper-12** | 0.535 | 0.268 | 0.480 |
| **LauraGPT (Ours)** | 0.507 | 0.312 | 0.492 |

**AAC Evaluation** We evaluate the AAC performance of LauraGPT on the evaluation set of Clotho (Drossos et al., 2020). Alongside the dataset, an attention-based encoder-decoder network (EncDec-Attn) is released and we cite its results as a baseline. We also compare LauraGPT with a competitive ensemble model Koizumi et al. (2020), which comprises an audio embedding model, a caption keyword estimation model, and a meta keyword estimation model. The evaluation is based on Huggingface's implementation, reporting BLEU-4, SPICE, CIDEr, and SPIDERr metrics. Five annotated captions are utilized to calculate the metrics for each test sample, and the results are shown in Table 6. We find that LauraGPT outperforms EncDec-Attn on all metrics. Compared to the ensemble model, LauraGPT achieves a comparable SPICE score while the ensemble model excels on CIDEr and SPIDEr. While SPICE is designed to capture the specificity of the generated captions, CIDEr focuses on evaluating the consensus between the generated captions and the reference captions. Thus, these results indicate that LauraGPT tends to generate captions that closely match one of the references, whereas the ensemble model prioritizes maintaining consensus of multiple references.

Table 6: Comparison of different models on the AAC task.

| Model | BLEU-4 ↑ | SPICE ↑ | CIDEr ↑ | SPIDEr ↑ |
|---|---|---|---|---|
| **Oracle** | 1.00 | 0.43 | 2.64 | 1.54 |
| **Drossos et al. (2020) (EncDec-Attn)** | 0.02 | - | 0.10 | - |
| **Koizumi et al. (2020) (Ensemble)** | - | 0.09 | 0.32 | 0.21 |
| **LauraGPT (Ours)** | 0.08 | 0.08 | 0.22 | 0.15 |

**SE Evaluation** We randomly select 500 clean utterances from the Librispeech test-clean set and mix them with noises from the test set of FSD50K at random SNRs of $[2, 5, 7, 10, 12, 15]$. We include the **SOTA** model on SE tasks as a baseline, which is a conformer-based metric generative adversarial network (CMGAN) Cao et al. (2022). To ensure a fair comparison, we train CMGAN[5] using the same SE training data as LauraGPT. Results are shown in Table 7. Compared to original noisy utterances (Noisy), LauraGPT improves speech quality (PESQ) and intelligibility (STOI), leading to a significant reduction in recognition error rates. This suggests that LauraGPT indeed enhances the noisy speech and reduces the acoustic interference. Compared to the SOTA CMGAN, LauraGPT achieves comparable PESQ scores, while CMGAN outperforms on STOI, CER, and WER. Considering that CMGAN incorporates multiple discriminators during the training stage, we hypothesize that incorporating adversarial losses and fine-tuning the GPT backbone with the codec vocoder in an end-to-end manner may help close the gap between LauraGPT and CMGAN.

**TTS Evaluation** We construct test samples using LibriTTS and AISHELL-1 corpora for Chinese and English TTS, respectively. VALL-E (Wang et al., 2023a) is **SOTA** zero-shot speaker adaptive TTS model. We re-implement two VALL-E models as competitive baselines with 0.34B trainable parameters, both trained with the same data as LauraGPT. One VALL-E model uses phonemes

---

[5]Use https://github.com/ruizhecao96/CMGAN

Table 7: Comparison of different models on the SE task.

| Model | PESQ ↑ | STOI ↑ | CER ↓ | WER ↓ |
|---|---|---|---|---|
| **Clean** | 4.50 | 100.0 | 3.31 | 7.55 |
| **Clean_codec_syn** | 2.72 | 87.0 | 7.46 | 14.28 |
| **Noisy** | 2.34 | 85.0 | 13.81 | 23.00 |
| **CMGAN** | 2.95 | 91.0 | 6.42 | 12.29 |
| **Discrete IO** | 1.96 | 64.0 | 40.91 | 53.97 |
| **LauraGPT (Ours)** | 2.97 | 88.0 | 9.05 | 15.94 |

(**Phone**) as the text input representation, while the other uses WordPiece tokens (**Token**) from the text tokenizer. Table 8 shows that VALL-E Phone outperforms VALL-E Token, indicating the importance of text representation for TTS task. Compared to both VALL-E models, LauraGPT achieves comparable speaker similarity and speech quality. The degradation in content consistency from LauraGPT results from the generalization issue, since the training data is too limited for the large LauraGPT model with 2B parameters.

Table 8: Comparison of VALL-E and LaruaGPT on Zero-Shot TTS task in terms of content consistency (CER/WER), speaker similarity (SECS), and speech quality (MOSNet).

| Model | AISHELL | | | LibriTTS | | |
|---|---|---|---|---|---|---|
| | CER ↓ | SECS ↑ | MOSNet ↑ | WER ↓ | SECS ↑ | MOSNet ↑ |
| **Origin** | 1.70 | 0.92 | 3.27 | 2.90 | 0.94 | 3.35 |
| **VALL-E Phone** | 4.75 | 0.91 | 3.22 | 4.30 | 0.92 | 3.28 |
| **VALL-E Token** | 6.52 | 0.91 | 3.19 | 6.57 | 0.93 | 3.28 |
| **LauraGPT (Ours)** | 6.91 | 0.90 | 3.14 | 8.62 | 0.91 | 3.26 |

## 5.2 DISCRETE VERSUS CONTINUOUS REPRESENTATIONS FOR AUDIO INPUTS

Different from existing unified audio-text models that use discrete tokens to represent audio inputs, LauraGPT employs the continuous features. We investigate the impact of discrete versus continuous representations for audio inputs by comparing results from LauraGPT to those from Discrete IO on ASR, S2TT, and SE tasks, which demonstrate capacity of audio recognition, understanding, and generation, respectively. We flatten the outputs of the first four quantizers to represent audio signals for both inputs and outputs, resulting in a token rate of 100 Hz. Table 2 shows that replacing continuous features with discrete audio tokens significantly degrades the ASR performance. For the S2TT task (Table 4), the model utilizing discrete tokens as inputs only yields BLEU scores of 5.1 and 5.0 on BSTC dev and CoVOST2 test sets, indicating lack of translation capability. Regarding the SE task (Table 7), using codec tokens as inputs cannot improve speech quality and intelligibility, which can be attributed to two reasons. Firstly, the distribution of noisy speech is too extensive to be effectively modeled by a single codec model trained only on clean speech. Secondly, the results in the "Clean" and "Clean_codec_syn" rows demonstrate that solely utilizing the first four groups to synthesize waveforms leads to degradation in all metrics. Note that "Clean_codec_syn" represents waveforms that are reconstructed using the first four codec groups extracted from clean speeches. "Clean_codec_syn" results can be considered as the upper bound for discrete token-based models.

## 6 CONCLUSION

In this paper, we propose LauraGPT, a novel and versatile GPT model for audio recognition, understanding, and generation. LauraGPT can handle both audio and text inputs and outputs, and perform a wide range of tasks related to content, semantics, paralinguistics, and audio-signal analysis. We demonstrate that LauraGPT can achieve competitive or superior performance compared to strong baselines across various audio processing benchmarks. We show that LauraGPT effectively combines both continuous and discrete features for audio, and benefits from supervised multitask fine-tuning. In future work, we plan to extend LauraGPT to instruction-following multi-modal foundation models with general comprehension and generation capabilities for text, speech, audio, and music.

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

# Appendices

## A   EXPERIMENTAL DETAILS

### A.1   BASIC TASKS

In our experiments, the following tasks are involved: **Automatic speech recognition (ASR)** is a vital task in the speech processing community. It focuses on transcribing speech into textual content. **Spoken language understanding (SLU)** is a task of deriving high-level semantic meaning directly from audio input. It aims to identify the user's intent and the relevant entity slots that fill the intent. An intent is composed of a scenario type and an action type, while slots and fillers are key-value pairs that specify the details of the intent. **Speech to text translation (S2TT)** is similar to machine translation, but it takes speech as input rather than text. **Speech emotion recognition (SER)** categorizes the emotions in speech input. Compared to textual emotion recognition, speech signals convey additional information, including tone and speed, which enhances emotion recognition. **Automated audio captioning (AAC)** aims to generate a natural language sentence that describes the content of an audio clip. **Speech enhancement (SE)** is an audio-to-audio task that aims to improve speech quality through noise suppression and dereverberation. In order to incorporate this task into a unified framework, we reformulate it as a classification problem using codec tokens. **Text-to-speech synthesis (TTS)** can be considered as the inverse process of ASR, where it generates speech that matches the given text. S2TT directly translates the source language speech into the target language text. **Machine translation (MT)** aims at translating text sequences from one language to another.

### A.2   TRAINING DATASETS

To ensure reproducibility, we exclusively utilize publicly available datasets to train and test LauraGPT. The training sets for the basic tasks in Section 3.5 are prepared as follows. For the ASR task, we utilize open-source Chinese datasets such as AISHELL-1 (Bu et al., 2017), AISHELL-2 (Du et al., 2018), WenetSpeech (Zhang et al., 2022), as well as open-source English datasets including LibriSpeech (Panayotov et al., 2015) and GigaSpeech (Chen et al., 2021a). For the S2TT task, we employ the commonly used BSTC (Zhang et al., 2021) and CoVOST 2 (Wang et al., 2020) datasets. Due to the limited data volumes of BSTC and CoVoST 2, we further augment the training set by translating AISHELL-1 and AISHELL-2 datasets into English and translating LibriSpeech dataset into Chinese using a publicly available text translation model (Wei et al., 2022b). Consequently, we obtain approximately 2,000 hours of supplementary data for Chinese-to-English and English-to-Chinese S2TT tasks. For the SER task, we collect corpora including MELD (Poria et al., 2018), IEMOCAP (Busso et al., 2008), RAVDESS (Livingstone & Russo, 2018), TESS (Pichora-Fuller & Dupuis, 2020), Crema-D (Cao et al., 2014), Emov-DB (Adigwe et al., 2018), and SAVEE (Jackson & Haq, 2014). These corpora are recorded in multi-modal formats, comprising audio or visual data. For the SLU task, we use the multi-domain Spoken Language Understanding Resource Package (SLURP) dataset (Bastianelli et al., 2020), which covers 18 scenarios. For the AAC task, we use AudioCaps (Kim et al., 2019), WavCaps (Mei et al., 2023), and Clotho (Drossos et al., 2020) datasets. For the SE task, pairs of noisy and clean speech are required for training. The clean utterances are extracted from the AISHELL-1, AISHELL-2, LibriSpeech, and WSJ datasets Paul & Baker (1992), while the noisy counterparts are generated by mixing the clean speech with noises from the FSD-50K dataset (Fonseca et al., 2022) at random signal-to-noise rates (SNR) ranging from 2 to 15. Besides the additional noises, we also simulate convolutional noises by convolving the clean speech data with room impulse responses (Ko et al., 2017). As a result, we obtain approximately 6000 hours of paired data for the SE task. For the TTS task, we use the open-source LibriTTS and 3D-speaker datasets (Zheng et al., 2023). For the MT task, we use the ParaCrawl v9 dataset (Kocmi et al., 2022), which consists of 14M parallel sentences for Zh→En and En→Zh translation. Further details of the training data for all tasks can be found in Table 9. For audio inputs, we extract the log-compressed Mel spectrogram features and feed them into the audio encoder (Section 3.3), while the audio outputs are discretized into tokens using the audio tokenizer (Section 3.2). As for the text data, both inputs and outputs are processed by the text tokenizer (Section 3.1).

Table 9: Statistics of the training data for basic tasks in Section 3.5. Corpus$^{\times N}$ means that the training samples in this corpus are copied $N$ times during training.

| Task | Training Data | # Samples |
|------|---------------|-----------|
| **ASR** | AISHELL-1, AISHELL-2, WenetSpeech, LibriSpeech, GigaSpeech | 24.2 M |
| **SLU** | SLURP$^{\times 10}$ | 1.2 M |
| **S2TT** | BSTC, CoVOST 2, AISHELL-1, AISHELL-2, LibriSpeech | 2.2 M |
| **SER** | MELD$^{\times 10}$, IEMOCAP$^{\times 10}$, RAVDESS$^{\times 10}$, TESS$^{\times 10}$ Crema-D$^{\times 10}$, Emov-DB$^{\times 10}$, SAVEE$^{\times 10}$ | 0.3 M |
| **AAC** | Clotho$^{\times 10}$, AudioCaps$^{\times 10}$, WavCaps$^{\times 5}$ | 1.3 M |
| **SE** | AISHELL-1, AISHELL-2, LibriSpeech, WSJ, FSD-50K, RIR | 5.3 M |
| **TTS** | LibriTTS$^{\times 2}$, 3D-Speaker$^{\times 2}$ | 5.0 M |
| **MT** | ParaCrawl | 14.2 M |

## A.3 EVALUATION DATASETS AND METRICS

Table 10 presents the evaluation datasets and evaluation metrics for various tasks. The metrics used in our experiments are described below:

- **CER** stands for Character Error Rate, a commonly used metric to evaluate the recognition performance of Chinese and English utterances. We also utilize CER to assess the content consistency in TTS task.
- **WER** stands for Word Error Rate, which considers entire words rather than individual characters. In our experiments, we use WER to evaluate ASR recognition performance, content consistency in TTS, and speech intelligibility in SE.
- **SECS**, which stands for Speaker Encoder Cosine Similarity, utilizes speaker embeddings extracted from a pre-trained speaker verification model [6] for both prompt and synthesized speech. The cosine similarity between the two embeddings is then employed to measure the speaker similarity between the prompt speech and the synthesized speech. Furthermore, the naturalness of the synthesized speech is evaluated using **MOSNet**, a non-intrusive score derived from a pre-trained neural network [7].
- **BLEU** and **BLEU-4** represent the Bilingual Evaluation Understudy metric and its extension, respectively, considering the precision of 4-grams. BLEU is commonly used to assess the quality of machine-generated text by comparing it to reference translations. In our experiments, we use BLEU to evaluate MT and S2TT, while the BLEU-4 extension is employed for AAC.
- **PESQ** represents Perceptual Evaluation of Speech Quality, while **STOI** stands for Short-time Objective Intelligibility. Both metrics are widely used to assess speech enhancement. PESQ ranges from $-0.5$ to $4.5$, whereas STOI is in the range of $[0, 1]$.
- **SPICE**, **CIDEr** and **SPIDEr** are metrics borrowed from the image captioning task and employed for AAC evaluation. SPICE stands for Semantic Propositional Image Caption Evaluation, CIDEr denotes Consensus-based Image Description Evaluation, and SPIDEr represents the average of SPICE and CIDEr.
- **WA**, **UA** and **WF1** stands for weighted accuracy, unweighted accuracy and the weighted F1 score. WA corresponds to the overall accuracy, UA corresponds to the average class-wise accuracy, and WF1 corresponds to the average class-wise F1 score.
- **ACC** measures the accuracy of predicting the scenario, action, and intent (i.e., the combination of scenario and action) in SLU evaluation. The **Word-F1**, **Char-F1**, and **SLU-F1** metrics assess the slot filling performance. Word-F1 and Char-F1 are based on the span-based F-measure. Word-F1

---

[6]Code is available at https://huggingface.co/microsoft/wavlm-base-plus-sv
[7]Code is available at https://github.com/lochenchou/MOSNet

uses word-level spans, while Char-F1 uses character-level spans. SLU-F1 is a metric that balances Word-F1 and Char-F1, computed as the sum of the confusion matrices.

Table 10: Evaluation datasets and metrics for different tasks. ↑ indicates that higher values of the metric are desirable, while ↓ implies the opposite.

| Task | Evaluation Datasets | Evaluation Metrics |
|------|---------------------|---------------------|
| **ASR** | AISHELL-1 test, AISHELL-2 test-ios, Librispeech test-clean & test-other | CER(↓), WER(↓) |
| **SLU** | SLURP test | ACC(↑), Word-F1(↑), Char-F1(↑), SLU-F1(↑) |
| **S2TT** | BSTC dev, En→Zh subset of CoVOST2 | BLEU(↑) |
| **SER** | MELD test | WA(↑), UA(↑), WF1(↑) |
| **AAC** | Clotho eval | BLEU-4(↑), SPIDEr(↑), CIDEr(↑), SPICE(↑) |
| **SE** | Librispeech test-clean, FSD50K, noise-92 | PESQ(↑), STOI(↑), WER(↓) |
| **TTS** | AISHELL-1 test, LibriTTS test-clean | CER(↓), WER(↓), SECS(↑), MOS(↑) |
| **MT** | WMT22 test | BLEU(↑) |

### A.4 DETAILS OF TRAINING SETUP

In all experiments, we optimize the model parameters through the following steps: (1) We initialize the Qwen backbone and audio encoder with the pre-trained checkpoints. (2) We then perform multi-task finetuning.

Due to the significant variation in data volume across different tasks, the training process is divided into three stages. In the first training stage, the model is fine-tuned on all tasks using the complete training as shown in Table 9. The AdamW optimizer is utilized with a peak learning rate of $5 \times 10^{-4}$ and 10K warmup steps. At the second stage, we further fine-tune the model on tasks that have small-scale datasets, including TTS, SE, AAC, SER, and SLU tasks. The AdamW optimizer is utilized with a peak learning rate of $2 \times 10^{-4}$ and 10K warmup steps. In the third training stage, we fine-tune the model on all tasks using the complete training set again. The peak learning rate of the AdamW optimizer for the third stage is reduced by half as $1 \times 10^{-4}$, while the warmup step remains at 10K.

For the codec vocoder, we train the predictor on the training data of the TTS and SE tasks. We use the Adam optimizer with a peak learning rate of 0.001 and 25K warmup steps. The decoder of the codec vocoder is initialized with the pre-trained checkpoints[8] and kept frozen during the multi-task finetuning of LauraGPT.

### A.5 DETAILS OF SER EVALUATION

During the training stage, emotion labels within different training corpora are unified into the following nine classes: anger, disgust, neutral, like, sadness, surprise, happiness, joy, and fear. At the test stage, we map the "like" and "happiness" emotion classes into the "joy" class to match the MELD test set. LauraGPT uses an autoregressive structure to generate emotion labels. Out-of-domain outputs are considered as classification errors, making the task harder. Both WavLM Base model and WavLM Large model utilize the weighted sum of multiple layers with learnable parameters as speech features, which are fed into a downstream network for classification.

### A.6 MT EVALUATION

LauraGPT can also support text-to-text tasks such as MT. Here we use WMT22 to evaluate the performance of Zh↔En MT task and the results are summarized in Table 11. We cite results

---

[8]https://funcodec.github.io

from Vega-MT (Zan et al., 2022) and HuaweiTSC (Wei et al., 2022a) in the WMT22 competition. Simultaneously, we take Transformer-Big (Zan et al., 2022) as the baseline, comprising 6 stacks with hidden nodes of 4096-1024. By employing the LauraGPT model for generating translations, we attain BLEU scores of 15.5 and 29.5 for Zh→En and En→Zh, respectively. Compared to the baseline, LauraGPT shows a notable decline in translation quality, which could potentially be attributed to the restricted proportion of the translation training data for LauraGPT. We only use the ParaCrawl dataset for MT training for LauraGPT, without implementing common strategies such as data augmentation. Despite the limited MT training data, LauraGPT still demonstrates its capability to fairly perform Chinese-English translation.

Table 11: Comparison of different models on MT task.

| Model | Zh→En | En→Zh |
|---|---|---|
| **Zan et al. (2022) Transformer-BIG** | 21.9 | 33.2 |
| **Zan et al. (2022) Vega-MT** | 33.5 | 49.7 |
| **Wei et al. (2022a) HuaweiTSC** | 29.8 | 49.7 |
| **LauraGPT** | 15.5 | 29.5 |

## B  ANALYSIS OF CRITICAL DESIGN CHOICES

### B.1  BATCH NORMALIZATION VERSUS LAYER NORMALIZATION IN AUDIO ENCODER

In the original design, batch normalization is applied after the convolution module in the Conformer-based audio encoder. However, we discover that this choice leads to endless looping decoding due to inaccurate estimations of mean and variance, particularly for tasks with long sequence lengths. When the issue of endless looping decoding occurs, the model generates several fixed tokens repeatedly and can't stop the generation until achieving a pre-defined maximum length. To address this issue, we replace batch normalization with layer normalization, which is more robust to various mini-batch sizes. We validate this design by focusing on the SE task, which generally has the longest sequence among all the included tasks. The results are shown in Table 12. To evaluate the occurring probability of endless loop decoding, we define the metric, "loop ratio", which represents the fraction of endless decoded cases among all test cases. The results indicate that batch normalization causes a significantly high loop ratio at the inference stage, leading to unacceptable PESQ and STOI scores. In contrast, **by replacing batch normalization with layer normalization, we observe a considerable reduction in the loop ratio to a very low level, thereby greatly improving the speech enhancement performance**. It should be noted that although the loop ratio of layer normalization is restricted, further research is still desired to explore more general normalization methods suitable for all audio-text tasks.

Table 12: Comparison of batch normalization and layer normalization on the SE task in terms of Loop Ratio (%), PESQ and STOI(%). ↑ indicates that higher values are desired, while ↓ implies the opposite.

| Normalization Type | Loop Ratio (↓) | PESQ (↑) | STOI (↑) |
|---|---|---|---|
| **Batch normalization** | 86.00 | 1.27 | 22.0 |
| **Layer normalization** | 4.60 | 2.97 | 88.0 |

### B.2  IMPACT OF INITIALIZATION FROM PRE-TRAINED MODELS

In LauraGPT, both the GPT backbone and audio encoder are initialized with the weights of pre-trained checkpoints. We investigate how the initialization affects the performance of LauraGPT. The experimental results for the ASR, S2TT and SE tasks are presented in Table 13, Table 14 and Table 15, respectively. From the results, we observe that the initialization has a significant impact on the performance of ASR and S2TT tasks, while its influence on the SE task is relatively limited. This suggests that the prior knowledge learned by the GPT backbone is crucial for text generation

tasks, but less important for audio generation tasks. Consequently, we hypothesize that **a reasonable approach to enhance the quality of generated audios could be to pre-train the GPT backbone not only with text sequences but also with audio token sequences**.

Table 13: Impact of initialization on the ASR task in terms of CER(%) ↓ for Chinese and WER(%) ↓ for English.

| Model | AISHELL-1 test | AISHELL-2 test-ios | Librispeech test-clean | Librispeech test-other |
|---|---|---|---|---|
| **LauraGPT** | 1.8 | 3.2 | 4.4 | 7.7 |
| **Without Init.** | 4.3 | 6.0 | 8.3 | 17.6 |

Table 14: Impact of initialization on the S2TT task in terms of BLEU.

| Model | Zh→En | En→Zh |
|---|---|---|
| **LauraGPT** | 17.8 | 38.5 |
| **Without Init.** | 8.4 | 12.2 |

Table 15: Impact of initialization on SE task in terms of Loop Ratio (%), PESQ and STOI(%). ↑ indicates that higher values are desired, while ↓ implies the opposite.

| Normalization Type | Loop Ratio (↓) | PESQ (↑) | STOI (↑) |
|---|---|---|---|
| **LauraGPT** | 4.60 | 2.97 | 88.0 |
| **Without init.** | 6.00 | 2.88 | 85.3 |

### B.3 EFFECTIVENESS OF MULTI-TASK FINETUNING

The multi-task learned single model of LauraGPT has the following advantages over individual single-task models: (1) Multi-task learning could potentially exploit synergy between related speech tasks and reduce over-fitting, hence LauraGPT could provide quality performance on a diverse set of tasks, and achieve better performance than single-task training, especially for tasks with limited training data. (2) Since multi-task learning could learn a single model capable of supporting a diverse set of tasks, it greatly simplifies the practical deployment and applications. LauraGPT can provide a diverse set of audio processing capabilities through the unified API and model implementation.

To further investigate the impact of multi-task finetuning and investigate whether multi-task learning could achieve better performance than single-task training, especially for tasks with limited training data, we conduct the following ablation experiments.

For the AAC, SLU, and SER tasks, which all suffer from limited training data, we compare their single-task performance with the multi-task trained LauraGPT. Specifically, we initialize the GPT with Qwen-LLM backbone and audio encoder with the same pre-trained checkpoints (i.e., the same initialization as LauraGPT before multi-task training), but train the model only using the AAC, SLU, and SER training data respectively. The results are shown in Table 16, 17 and 18.

For the AAC task, the evaluation set of Clotho is used for comparision. From Table 16, we find that the multi-task trained LauraGPT outperforms the single-task model on all metrics.

The similar results can be observed on the SLU task. As shown in Table 17, evaluated on the SLURP test set, the multi-task trained LauraGPT outperforms the single-task model on scenario/action/intent accuracy with **+1.89/+3.11/+2.88** absolute gains, especially with **+21.22/+23.83/+22.46** absolute gains on slot filling Word-F1/Char-F1/SLU-F1.

For the SER task, the MELD test set is selected. As shown in Table 18, for UA and WF1 metrics, multi-task trained LauraGPT can significantly outperforms the single-task model, while the WA result is slightly worse. To further analyze these results, we conduct a statistical analysis of the number of samples for each emotion class in both training and testing sets of the MELD dataset,

Table 16: Comparison of single-task finetuning and multi-task finetuning on the AAC task.

| Model | BLEU-4 ↑ | SPICE ↑ | CIDEr ↑ | SPIDEr ↑ |
|---|---|---|---|---|
| **Single-task** | 0.06 | 0.07 | 0.16 | 0.11 |
| **LauraGPT** | 0.08 | 0.08 | 0.22 | 0.15 |

Table 17: Comparison of single-task finetuning and multi-task finetuning on the SLU task.

| Model | Scenario (ACC ↑) | Action (ACC ↑) | Intent (ACC ↑) | Word-F1 ↑ | Char-F1 ↑ | SLU-F1 ↑ |
|---|---|---|---|---|---|---|
| **Single-task** | 89.15 | 85.96 | 84.99 | 49.98 | 52.03 | 50.99 |
| **LauraGPT** | 91.04 | 89.07 | 87.87 | 71.20 | 75.86 | 73.45 |

as well as their corresponding test accuracy. The results are shown in Table 19. Compared to single-task model, multi-task trained LauraGPT results in degradation in accuracy for classes with a larger number of training samples, while significantly improving accuracy on classes with fewer training samples. This explains why WA decreases slightly from multi-task training, but UA and WF1 show significant improvements. Note that WF1 is the primary metric on the MELD dataset due to the sample imbalance across different emotion classes Chen et al. (2023). Furthermore, the accuracy of the disgust and fear classes is 0, which aligns with the fact that these two classes have the fewest training samples in the MELD dataset. Multi-task training not only remarkably improves the performance of emotion classes with low accuracy (joy, sadness, surprise), but also greatly improves the performance of classes that cannot be predicted with single-task training (disgust, fear).

In summary, the ablation analysis results verify that multi-task learning of LauraGPT achieves better performance than single-task training for tasks with limited training data.

Table 18: Comparison of single-task finetuning and multi-task finetuning on the SER task.

| Model | WA ↑ | UA ↑ | WF1 ↑ |
|---|---|---|---|
| **Single-task** | 0.508 | 0.221 | 0.426 |
| **LauraGPT** | 0.507 | 0.312 | 0.492 |

Table 19: The Accuracies of single-task finetuning and multi-task finetuning for different emotion classes on the SER task.

| Model | anger | disgust | neutral | joy | sadness | surprise | fear |
|---|---|---|---|---|---|---|---|
| **#Training Samples** | 1109 | 271 | 4710 | 1743 | 683 | 1205 | 268 |
| **#Testing Samples** | 345 | 68 | 1256 | 402 | 208 | 281 | 50 |
| **Single-task** | 0.396 | 0.000 | 0.875 | 0.119 | 0.029 | 0.128 | 0.000 |
| **LauraGPT** | 0.333 | 0.103 | 0.708 | 0.381 | 0.236 | 0.381 | 0.040 |

