# OpenReview forum: "LAURAGPT: LISTEN, ATTEND, UNDERSTAND, AND REGENERATE AUDIO WITH GPT"
_ICLR.cc/2024/Conference — Submitted to ICLR 2024_

### Official Review · Reviewer_ncjQ · 2023-10-29

**Soundness:** 3 good
**Presentation:** 3 good
**Contribution:** 2 fair
**Rating:** 5
**Confidence:** 4

**Summary:**

In general, the paper is an experiment oriented work which demonstrates the GPT-style structure can do various speech tasks. Specifically, this paper introduces LauraGPT, a versatile GPT model designed for audio tasks, including: automatic speech recognition, speech-to-text translation, text-to-speech synthesis, machine translation, speech enhancement, automated audio captioning, speech emotion recognition, and spoken language understanding. To enable these capabilities, the model combines continuous and discrete audio features, utilizing an audio encoder for input and a discrete codec for output. The model is then fine-tuned through supervised multitask learning on a range of audio-to-text, text-to-audio, audio-to-audio, and text-to-text tasks. Extensive experiments demonstrate that LauraGPT achieves competitive or superior performance compared to existing state-of-the-art models across various audio processing benchmarks.

**Strengths:**

The paper presents a thorough set of experiments, showcasing the capabilities of LauraGPT in handling both audio and text inputs and generating outputs across a diverse range of tasks. These tasks encompass content analysis, semantics, paralinguistics, and audio-signal analysis.

As far as my knowledge extends, the paper provides extensive coverage of speech tasks in its evaluation, as indicated by the authors in Table 1.

**Weaknesses:**

While the paper presents solid research, it falls short in paving the way for future investigations.

1. By the end of 2023, speech researchers generally believe that GPT-style models can handle various speech tasks, even though certain specific tasks may not achieve state-of-the-art performance when compared to baseline models of the same size. Instead of offering insights beyond the extensive experiments conducted, the authors primarily focus on demonstrating the effectiveness of GPT-style models across multiple speech tasks.
I hope the authors will consider demonstrating whether multi-task learning can result in task synergy, where tasks can benefit from each other rather than being treated as separate or even conflicting objectives. For instance, if I have a 1B model that solely focuses on automatic speech recognition (ASR), would it outperform a 1B model capable of performing ASR, text-to-speech (TTS), and speech-to-text (ST) tasks? If the answer is no, then why should we incorporate all these tasks into the same model? It would be valuable for the authors to analyze the relationship between task performance in the context of multi-task learning, as evidenced in Table 2, where LauraGPT lags significantly behind the state-of-the-art LibriSpeech despite its 2B model size.

2. In the realm of fundamental speech models, are there any emerging points of interest similar to those in the field of natural language processing (NLP)? Exploring this aspect could be a valuable research direction. If speech researchers are unable to answer this question, I believe that running multi-task learning experiments alone may not be sufficient to construct the next generation of speech models.

**Questions:**

Apart from Speech translation, could you list more complex tasks that should use foundamental model to solve rather than do them one by one?

If the performance is worse than train an ASR model alone, what is the value of the multi-task learning model?

---

> ### Author Response · Authors · 2023-11-18
> **Responses to Reviewer ncjQ (Part 1/2)**
>
> We would like to thank the reviewer for all the constructive feedback. Below we address all of your questions and concerns.
>
> Please check our Global Responses to All Reviewers.
>
> $$ $$
>
> > Apart from speech translation, could you list more complex tasks that should use fundamental model to solve rather than do them one by one?
>
> **Response:**  As stated in Section 3.5, with its modular and flexible design, LauraGPT provides an extensible framework to support complex tasks. By breaking a task into sub-tasks among the basic tasks used in training and cascading the raw inputs and model outputs of sub-tasks, LauraGPT can perform more complex tasks than the basic tasks.
>
> Similar to the speech-to-speech translation (S2ST) example, LauraGPT can perform more complex tasks by chaining together basic tasks as described above. Here are a few examples of other complex tasks that LauraGPT can support rather than doing them one by one:
>
> 1. **Rich transcription**: We can extend LauraGPT to simultaneously transcribe audio into content, speaker information (speaker identification, etc), paralinguistic information (emotion, etc.) and high-level semantic information (intent, slots, etc.) by including different task IDs at the generation process. This approach could avoid error accumulation in a pipelined approach and is more efficient than performing these tasks individually.
>
> 2. **Rich translation**: We can further chain the above basic tasks of rich transcription with the MT task to perform rich translation of audio input. The audio input can provide additional paralinguistic information to improve the translation accuracy.
>
> 3. **Noise-robust ASR**: We can implement noise-robust ASR by chaining tasks and creating the following input sequence: [noisy speech embedding, <SE>, embedding of the enhanced speech,  <ASR>].  Since SE and ASR are jointly trained for LauraGPT, LauraGPT could effectively exploit embeddings of the original noisy speech and enhanced speech for noise-robust ASR.
>
> $$ $$
>
> > what is the value of the multi-task learning model ? It would be valuable for the authors to analyze the relationship between task performance in the context of multi-task learning
>
> **Response:** The multi-task learned single model of LauraGPT has the following advantages over single-task models:
>
> 1. Multi-task learning could potentially exploit the synergy between related speech tasks and reduce over-fitting, hence LauraGPT could provide quality performance on a diverse set of tasks, and achieve better performance than single-task training, especially for tasks with limited training data.
>
> 2. Since multi-task learning could learn a single model capable of supporting a diverse set of tasks, it greatly simplifies the practical deployment and applications. LauraGPT can provide a diverse set of audio processing capabilities through the unified API and model implementation.
>
> Please refer to Our Global Response to Q3 for more details. Our Global Response to Q3 clearly demonstrates the significant performance improvements from multi-task learning of LauraGPT over single-task training on tasks with limited training data.
>
> $$ $$
>
> > as evidenced in Table 2, where LauraGPT lags significantly behind the state-of-the-art LibriSpeech despite its 2B model size.
>
> **Response:** In our Global Response to Q1, we provide a summary of the performance comparisons between our multi-task learned LauraGPT with baselines on each task.  As analyzed below, on ASR task, LauraGPT produces competitive performance on  both Chinese and English test sets compared to competitive baselines.
>
> **For ASR task**,  the baselines are competitive Paraformer and Whisper Large V2.  As shown in Table 2,  on the Chinese test sets,  LauraGPT greatly outperforms Whisper by **-3.9** and **-2.3** absolute on CER and performs comparably to Paraformer with a much smaller amount of training data. On the English test sets, LauraGPT achieves comparable performance to Paraformer and performs better on the more noisy test set, but performs worse than Whisper Large V2 as Whisper Large V2 uses much more English training data than LauraGPT. Note that there are other **targeted optimized models** [1] that can achieve better performance than Whisper Large V2 on LibriSpeech test sets. However, they only focus on English speech recognition and benefit from additional technologies such as language model decoding.
>
> [1] C. S, et al. Wavlm: Large-scale self-supervised pre-training for full stack speech processing. IEEE Journal of Selected Topics in Signal Processing.

---

> ### Author Response · Authors · 2023-11-18
> **Responses to Reviewer ncjQ (Part 2/2)**
>
> > "In the realm of fundamental speech models, are there any emerging points of interest similar to those in the field of natural language processing (NLP)? Exploring this aspect could be a valuable research direction. If speech researchers are unable to answer this question, I believe that running multi-task learning experiments alone may not be sufficient to construct the next generation of speech models. " Thoughts and plans on the next generation of fundamental speech models.
>
> **Response:**  Thank you for the valuable question. There are emerging points of interest for fundamental speech models that are similar to those in the field of NLP.  This is a tremendously valuable research direction. Our work in this paper achieves one important milestone for this research question, as we explored and provided promising answers to the following question:
> - How to design more efficient and scalable GPT-style models than existing approaches that can leverage large-scale labeled data and achieve state-of-the-art performance on a diverse set of speech tasks including speech recognition, understanding and generation, using a single model?  Note that previous fundamental speech models either focus solely on speech recognition and understanding but neglect generation tasks, or support speech generation but suffer from significant performance degradation on recognition and understanding tasks.
>
> As shown in Appendix B.3 and Section 5.2, multi-task learning and the combination of continuous features and discrete features for audio is crucial for achieving the above goal.
>
> For the next generation of fundamental speech models, we are inspired by the recent advances of large language models (LLMs) in NLP, and we envision that the fundamental speech models should have the following capabilities:
> - In-context learning ability like GPT-3, which can learn from few-shot examples and adapt to new tasks, such as predicting the age of the speaker from a speech sample.
> - Instruction-following ability like InstructGPT and ChatGPT, which can perform the appropriate speech-related task given a natural language instruction, such as synthesizing a speech with a specific emotion or style.
> - General audio modeling ability, i.e., speech, non-speech audio, and music, such as music generation.
>
> We consider the following steps to progress from LauraGPT into the next generation of fundamental speech model:
> 1. Self-supervised pre-training exploring large-scale unlabeled data and enhancing cross-modality alignment. We plan to explore tasks such as next token prediction on unlabeled speech and text before the multi-task supervised training in our paper.
> 2. Extend the task categories for supervised pre-training, such as music understanding and generation, speaker tasks such as speaker diarization and so on. We plan to add more supervised training tasks into LauraGPT.
> 3. Construct instruction-following data and conduct supervised fine-tuning. We could generate instruction texts for all audio-related tasks using competitive LLMs (e.g., GPT-4) and pair the instruction with corresponding samples.
> 4. Optionally, we can collect data and conduct RLHF to make the fundamental speech model more harmless, helpful, and honest.
>
> Please note that our paper demonstrates that our current LauraGPT has made solid progress and reached one important milestone toward a speech foundation model. As demonstrated in the above steps, from LauraGPT to the next-generation speech foundation model we envisioned, **most remaining efforts are in more data collection and more training**.  **There is no need to modify the model architecture**. We hope that our paper can provide insights and inspiration for future research on fundamental speech models.

---

### Official Review · Reviewer_x7eV · 2023-11-02

**Soundness:** 3 good
**Presentation:** 2 fair
**Contribution:** 3 good
**Rating:** 6
**Confidence:** 3

**Summary:**

- LauraGPT is a single GPT-like LLM that operates on a combination of discrete and continuous features for audio signals and text, and is fine-tuned to perform a wide range of speech and audio processing tasks.
- A pre-trained text-only language model (Qwen) serves as the backbone for Laura. For audio, LauraGPT uses a combination of discrete tokens obtained from an improved Encodec-based audio codec (where only the first quantizer is used as the tokenizer), as well as a conformer-based encoder which is initialized with weights from a pertained ASR model. The autoregressive model predicts the next token (text or audio) given the input embeddings, task embeddings, and the previously predicted tokens. The output text is obtained from the Qwen tokenizer and the final audio is obtained from a so-called codec vocoder.
- Instead of directly using the decoder of the pre-trained audio codec, LauraGPT uses a codec vocoder wherein a transformer model serves to predicted the sum of all quantizers embedding for ground truth audio given just the first quantizer embedding and additional context. Subsequently, during inference, the predicted audio token embedding can be transformed into the summed token embeddings and passed to the pre-trained codec decoder to generated raw audio.
- The authors evaluate LauraGPT against strong baselines for each of the task it is capable of performing. LauraGPT performs well on most tasks, only failing to beat baselines in the Speech enhancement and TTS task. It also fails to beat Whisper-Large V2 in English, which is understandable given the smaller amount of English data it was pre-trained on.

**Strengths:**

- The paper demonstrates a strategy to fine-tune existing LLMs trained only for text to perform various audio processing (generation and understanding) tasks.
- Unlike other related work, this paper shows that utilizing a mix of continuous and discrete representations of audio in the transformer architecture leads to improved performance in the final generation task (as ablated in section 5.2).
- The evaluation is pretty comprehensive and strong baselines have been chosen for most tasks.

**Weaknesses:**

- The ablation regarding discrete tokens vs continuous + discrete tokens feels incomplete without also using the VALL-E style token prediction setup. Currently, the token prediction scheme is similar to that used by SPEAR-TTS wherein each quantizer level token is predicted one-by-one before moving on to the next audio-frame’s tokens.
- Some statements are not clearly backed up by experiments. For example, one of the main contributions listed is the fact that continuous and discrete representations of audio are used in LauraGPT and that this preserves both fidelity and generality of audio data. Firstly, I am not quite sure what it means to preserve generality of audio. Second, while it is shown in the ablation that the Discrete IO model suffers, it is not clear to me how these results show that fidelity and generality is preserved because of the use of combined representations. All I see is that performance on various task is improved by using the combination. Also, one additional benefit the combined representation model sees is the use of the codec vocoder. Perhaps that is the source of the improvements in LauraGPT?
- Section 3.4 would be well served with some more detail. The reader would benefit from some repeating information that the GPT model only uses the first quantizer. I found it difficult to understand initially and had to read from the start of section 3 again.
- A few figures going into more detail for each of the components in figure 1 would also greatly improve the readability of the the method section. Currently, figure 1 is very high-level and does not offer the reader too much.

**Questions:**

- Do I understand correctly that the model uses continuous features from only the input audio, and uses audio token embeddings for previous audio tokens, meaning that the generated audio is always seen as tokens within the GPT model? It would benefit the reader if this is stated in the text explicitly as well.

---

> ### Author Response · Authors · 2023-11-18
> **Responses to Reviewer x7eV (Part 1/2)**
>
> We would like to thank the reviewer for all the constructive feedback. Below we address all of your questions and concerns.
>
> Please check our Global Responses to All Reviewers.
>
> $$ $$
>
> > The ablation regarding discrete tokens vs continuous + discrete tokens feels incomplete without also using the VALL-E style token prediction setup. Currently, the token prediction scheme is similar to that used by SPEAR-TTS wherein each quantizer level token is predicted one by one before moving on to the next audio-frame’s tokens.
>
> **Response:**  Our response to this question is two-fold:
>
> 1. The goal of the current ablation study is to evaluate the impact of combining continuous features and discrete tokens in LauraGPT on **ALL** relevant spoken language processing tasks, where the inputs of GPT are continuous representations and the outputs are discrete tokens. The token prediction scheme of VALL-E can be applied to audio generation tasks, such as TTS and SE. However, it is not straightforward to apply it to **audio recognition and understanding tasks**, such as ASR, S2TT, and AAC. Therefore, we have not included the comparison to the VALL-E style token prediction setup in the paper.
>
> 2. On the other hand, taking SE as an example of audio generation tasks, we evaluated the VALL-E style token prediction in our preliminary experiments. In these experiments, we kept the model inputs and architectures the same as LauraGPT and only replaced the token prediction scheme with the VALL-E style. The results are as follows.  We find that the token prediction scheme of LauraGPT significantly outperforms VALL-E in terms of CER and WER and also improves PESQ while obtaining the same STOI. These results indicate the superiority of predicting discrete tokens through GPT and converting them into a continuous format using a codec vocoder.
>
> |Token prediction scheme|PESQ|STOI (%)|CER|WER|
> |------|------|-------|------|-----------------|
> |VALL-E|2.55|88.0|10.52|19.32|
> |LauraGPT|2.97|88.0|9.05|15.94|
>
> $$ $$
>
> > Some statements are not clearly backed up by experiments. For example, one of the main contributions listed is the fact that continuous and discrete representations of audio are used in LauraGPT and that this preserves both fidelity and generality of audio data. Firstly, I am not quite sure what it means to preserve generality of audio. Second, while it is shown in the ablation that the Discrete IO model suffers, it is not clear to me how these results show that fidelity and generality is preserved because of the use of combined representations. All I see is that performance on various task is improved by using the combination. Also, one additional benefit the combined representation model sees is the use of the codec vocoder. Perhaps that is the source of the improvements in LauraGPT?
>
> **Response:**  In Section 1, we explain that we use continuous features to represent the input audio to ensure high performance on **audio recognition and comprehension tasks**, and use codec-based discrete features to represent the output audio, which enables joint autoregressive modeling with text features for **audio generation tasks**. We show in our experiments that continuous features for audio (such as Filterbank) have notable advantages over discrete units on audio recognition, understanding, and audio-signal-related tasks, such as ASR, S2TT, and SE, as analyzed in Section 5.2. Therefore, our model achieves a better trade-off between model uniformity (generality) and high performance (fidelity) on diverse categories of audio tasks than existing approaches that use discrete features for both input and output audio. The generality of audio here refers to the ability of our model to handle different types of audio tasks in a unified framework, by using codec-based discrete features to represent the output audio and jointly modeling them with discrete text tokens. We attribute the fidelity of audio mainly to the use of continuous features for input audio. In the ablation experiment using Discrete IO in Section 5.2, we compare the performance of our LauraGPT with a model that uses discrete features for both input and output audio (Discrete IO). We find that Discrete IO significantly degrades ASR, S2TT and SE tasks, while LauraGPT has good performance on these tasks. This demonstrates that our model preserves both the fidelity and generality of audio data by using a combination of continuous and discrete representations.

---

> ### Author Response · Authors · 2023-11-18
> **Responses to Reviewer x7eV (Part 2/2)**
>
> > Do I understand correctly that the model uses continuous features from only the input audio, and uses audio token embeddings for previous audio tokens, meaning that the generated audio is always seen as tokens within the GPT model? It would benefit the reader if this is stated in the text explicitly as well.
>
> **Response:** Your understanding is correct. As stated in Section 3, for audio inputs, we extract the log-compressed Mel spectrogram features and feed them into the audio encoder, while the audio outputs are discretized into tokens using the audio tokenizer. This means that the generated audio is always seen as tokens within the GPT model, as you correctly understood. We have made this point more explicit in the revised version.
>
> $$ $$
>
> > Section 3.4 would be well served with some more detail. The reader would benefit from some repeating information that the GPT model only uses the first quantizer. I found it difficult to understand initially and had to read from the start of section 3 again.
>
> **Response:** We have clarified the details of only using the first quantizer to tokenize audio signals in the revised Section 3.4. The remaining 31 quantizers are only used at the training stage of the codec vocoder.

---

### Official Review · Reviewer_Xm2a · 2023-11-07

**Soundness:** 3 good
**Presentation:** 3 good
**Contribution:** 3 good
**Rating:** 6
**Confidence:** 5

**Summary:**

In this paper, the authors propose a unified GPT model(LauraGPT) for audio recognition, understanding, and generation. They encode input audio into continuous representation and decode output audio from discrete codec codes and fine-tune a language model. They evaluate the LauraGPT on various audio processing benchmarks like ASR, S2TT, TTS and so on. The experimental results conducted on tasks show the effectiveness of the LauraGPT and the flexible design of the model.

**Strengths:**

1.The proposed model supporting largest number of and most diverse audio processing tasks compared with other structure, which is interesting and reasonable. The authors also give detailed analysis and descriptions about these tasks and results with baselines.

2.The article provides a clear categorization of tasks and the model provides an extensible framework to support complex tasks with its modular and flexible design. It can break a task into sub-tasks among the basic types and perform well. This makes the model well extensible.

3.The model combines continuous and discrete features for audio signals. It utilizes the continuous features and analyzes the impact of discrete versus continuous representations in ASR, S2TT, and SE tasks.

**Weaknesses:**

1.The task-related token included in the matrices is not explained enough, how is it utilized and how is it embedded to give the information of the types of the tasks. It lacks some details about it in the description.

2.In evaluation part, there’s a lack of adequate analysis of the relationship between the poor performance in some tasks and model size.

3.In Part 3, there is a lack of detailed visualizations to show the internal framework of the model, as well as the details of the training and inference process.

**Questions:**

1.In the article, the model is able to perform in more task domains, compared to the most related multi-task unified audio-text models, but in the comparison, why is there no comparison with these multi-task models for the various metrics of these tasks?

---

> ### Author Response · Authors · 2023-11-18
> **Responses to Reviewer Xm2a**
>
> We would like to thank the reviewer for all the constructive feedback. Below we address all of your questions and concerns.
>
> Please check our Global Responses to All Reviewers.
>
> $$ $$
>
> > The task-related token included in the matrices is not explained enough, how is it utilized and how is it embedded to give the information of the types of the tasks. It lacks some details about it in the description.
>
> **Response:** The task-related token (also called taskid in the paper) is added as a special symbol directly to the dictionary. Therefore, it can be converted to an embedding through an embedding matrix like the text tokens. For example, in the case of the ASR task, the input audio is passed through an audio encoder to obtain the corresponding audio embedding. The task-related token and the output text are converted to the text embedding through the embedding matrix, where the first embedding corresponds to the <ASR> taskid. These embeddings are then concatenated as [audio embedding, text embedding], which serves as the input for QWen LLM.
>
> The task-related token can help the model distinguish between different tasks with the same input. We use ASR and S2TT tasks as an example. Given the same input speech utterance, if we add <ASR> task ID after the audio embedding, the model will generate the ASR recognized result. And if we add <S2TT> taskid, the model will output the translated text.
>
> $$ $$
>
> > In evaluation part, there’s a lack of adequate analysis of the relationship between the poor performance in some tasks and model size.
>
> **Response:** Please refer to our Global Response to Q1 for performance analysis on all tasks, which includes a detailed analysis of reasons causing relatively lower performance of LauraGPT on some metrics on some test sets. Note that none of the relatively lower performance of LauraGPT compared to baselines is caused by model size differences.
>
> $$ $$
>
> > All the rest concerns under Weaknesses and Questions
>
> **Response:**  Please refer to our Global Responses to All Reviewers.

---

### Official Review · Reviewer_Nqbz · 2023-11-08

**Soundness:** 3 good
**Presentation:** 3 good
**Contribution:** 3 good
**Rating:** 5
**Confidence:** 4

**Summary:**

This paper describes an integrated Audio-Text LLM that uses continuous features to represent input audio and discrete tokens to generate output audio. This allows it to be used for audio generation and audio->audio tasks like text-to-speech synthesis and speech enhancement, in contrast to most (but not all) existing Audio-Text systems. The system is evaluated on many standard tasks in the various modalities and in comparison to reasonable existing single-task systems provides significantly improved performance on spoken language understanding accuracy, speech to text translations from english to Chinese, equivalent performance on ASR, spoken language understanding f1, speech emotion recognition, speech to text translation from Chinese to english, and worse performance on automatic audio captioning, speech enhancement, and text-to-speech. Overall, the system seems competitive in these various tasks compared to these baselines.

**Strengths:**

* The problem of general audio-text understanding, modeling, and generation is an important one, as is multi-modality in LLMs in general.
* Ambitious combination of many tasks into a single model
* Experiments seem well conducted, reasonable selection of tasks and benchmarks

**Weaknesses:**

In terms of novelty, this is a popular area of research at the moment, with SeamlessM4T being released in August, a few weeks before the submission deadline in addition to the many relevant recent references cited here. If the new capability enabled by the proposed approach is audio output, then one of the most relevant systems appears to be AudioGPT, which also supports speech enhancement. The reason that it is not compared or included in Table 1, which shows capabilities of different existing models, is that it integrates an "expert audio model with LLM", which I don't fully understand the meaning of. An expanded explanation of this would be very useful in understanding this key point.

Also regarding Table 1, it should be clarified that it is showing just what the model has been trained/evaluated on, not what it is necessarily capable of. For example, any system that can perform speech to text translation should also be able to perform automatic audio captioning. Similar arguments hold for speech emotion recognition and spoken language understanding.

In terms of the significance of the results, it is interesting that these tasks can all be solved by a single model, but it's not clear that doing so gives the model advantages over separate models. The performance seems mostly on par. It is also not clear whether the other multimodal systems described in Table 1 (especially SpeechT5) would do better than the proposed system on these tasks as the comparisons are only against single-task systems trained on much less data (the subset of the data that the proposed system was trained on for each particular task).

In terms of clarity, two different taxonomies of related models are introduced in sections 1 and 2, these could be combined into a single one to make space for more explanation of the data that the model was trained on from the appendix. In particular, it is not clear in the body of the paper whether the model is trained once on all of the data or separately for different tasks or how that is navigated and how much data it is overall.

Some claims about the proposed model's superiority are not well supported by the results. Specifically the claim of being best on SLU, when it is really just SLU accuracy, but not f1 scores. This is also the case for SER in that the proposed model is better on unweighted accuracy, but not weighted f1 or accuracy.

Minor comments:
* A definition of the "endless looping problem" and "loop ratio" would be helpful in the appendix
* "These results indicate that LauraGPT tends to generate captions that closely match one of the references..." can you explain how you reach this conclusion?
* Please define exactly what "clean_codec_syn" is in table 7
* In the appendix, prosody includes both tone and speed, so no need to list them separately
* In the appendix, I believe "dereverberation" is meant instead of "echo cancellation" which involves an echo back to the far end of a telephone call, typically.
* In the appendix, "For the SER task, we collect corpora including..." are there other corpora used? If so, please list them. If not, reword.

**Questions:**

Can you clarify the difference between the proposed system and AudioGPT?

---

> ### Author Response · Authors · 2023-11-18
> **Response to Reviewer Nqbz (Part 1/2)**
>
> We would like to thank the reviewer for all the constructive feedback. Below we address all of your concerns and questions.
>
> Please check our Global Responses to All Reviewers.
>
> $$ $$
>
> >One of the most relevant systems appears to be AudioGPT, but it is not compared in Table 1. Explain expert audio model with LLM. Can you clarify the difference between the proposed system and AudioGPT?
>
> **Response:**  We respectfully disagree on considering AudioGPT as one of the most relevant systems to our LauraGPT. As we stated in Section 2.1, our LauraGPT is a **single unified audio-text model that can directly process both audio and text inputs and generate outputs in speech and text modalities**. In contrast,  AudioGPT is an integrated AI system that **integrates and interfaces specialized audio models, such as ASR and TTS models,  for speech input and speech output (i.e., the expert audio model we mentioned in Section 2.1) with LLM**. Note that these specialized audio models add more complexity, consume more resources, and cause AudioGPT to be prone to unavoidable error accumulation problems.  Hence, AudioGPT is drastically different from our LauraGPT and is not compared to or included in Table 1.
>
> $$ $$
>
> >Regarding Table 1, it should be clarified that it is showing just what  the model has been trained/evaluated on, not what it is necessarily  capable of.
>
> **Response:** We agree with you that Table 1 only shows the tasks the most related multi-task unified audio-text models are trained and evaluated on, not their capabilities. We have updated the caption of Table 1 to clarify this and also revised the text in Section 1 since VioLA and AudioPaLM should be able to perform AAC. However, it is important to point out that decoder-only models using discrete speech tokens such as VioLA and AudioPaLM may suffer from the information loss caused by quantization of speech signals into discrete tokens, which leads to significant performance degradation over models using continuous speech features, as our ablation study in Section 5.2 shows that by using continuous features for audio input, LauraGPT significantly outperforms the counterpart using discrete features on ASR, S2TT, and SE tasks.
>
> $$ $$
>
> >All other questions and Weaknesses.
>
> **Response:**  Please refer to our Global Responses to All Reviewers.

---

> ### Author Response · Authors · 2023-11-18
> **Response to Reviewer Nqbz (Part 2/2)**
>
> **Minor Comments:**
>
> > A definition of the "endless looping problem" and "loop ratio" would be helpful in the appendix
>
> **Response:** For some test cases of SE task, if the model generates several fixed tokens repeatedly and cannot stop the generation until reaching a pre-defined maximum length, we consider the problem of endless looping decoding occurs and consider this test case to be an "endless decoded case". To evaluate the probability of occurrence of this problem, we define the metric, "loop ratio", which refers to the fraction of "endless decoded cases" among all test cases. We have clarified the definition of "endless looping problem" and "loop ratio" in the revised Appendix B.1.
>
> $$ $$
>
> >"These results indicate that LauraGPT tends to generate captions that closely match one of the references..." can you explain how you reach this conclusion?
>
> **Response:** As explained in our Global Response to Q1, for the ACC task, SPICE is designed to capture the **specificity** and accuracy of the generated captions in terms of the semantic content, while CIDEr focuses on evaluating the **consensus** or agreement between the generated captions and the reference captions. SPIDEr can be viewed as an average of SPICE and CIDEr. From Table 6, we find that the single model LauraGPT achieves a comparable SPICE score but underperforms on CIDEr and SPIDEr, compared with the Ensemble baseline. Therefore, we think that LauraGPT tends to generate captions that closely match one specific reference rather than the consensus of references. We have clarified this reasoning process in the AAC Evaluation paragraph in Section 5.1 of the revised paper.
>
> $$ $$
>
> > Please define exactly what "clean_codec_syn" is in table 7
>
> **Response:** In Table 7, "Clean_codec_syn" refers to waveforms that are reconstructed using the first four codec groups extracted from the clean speeches. The results corresponding to  "Clean_codec_syn" can be considered as the upper bound for the Discrete IO models. The results in the "Clean" and "Clean_codec_syn" rows demonstrate that solely utilizing the first four codec groups extracted from the clean speeches to synthesize waveforms leads to degradation in speech quality (PESQ), intelligibility (STOI), and a significant degradation in recognition error rates. These results demonstrate the superiority of our choice of continuous representations for audio inputs. We have added the definition of "Clean_codec_syn" and this clarification in Section 5.2 in the revised paper.
>
> $$ $$
>
> > In the appendix, prosody includes both tone and speed, so no need to list them separately
>
> **Response:** Thank you for the advice. We have removed the word "prosody" and listed "tone" and "speed" to specify the additional information conveyed in speech but not in the text, in the revised paper.
>
> $$ $$
>
> > In the appendix, I believe "dereverberation" is meant instead of "echo cancellation" which involves an echo back to the far end of a telephone call, typically.
>
> **Response:** Thank you for the advice.  We have changed the term "echo cancellation" to "dereverberation" in the revised version.
>
> $$ $$
>
> > In the appendix, "For the SER task, we collect corpora including..." are there other corpora used? If so, please list them. If not, reword.
>
> **Response:**   No other corpora are used for the SER task. We have also modified this description in the revised paper.
>
> $$ $$
>
> > In particular, it is not clear in the body of the paper whether the model is trained once on all of the data or separately for different tasks or how that is navigated and how much data it is overall.
>
> **Response:** As described in Section 4 of the paper, we initialize the Qwen backbone and audio encoder with the pre-trained checkpoints and then optimize the model parameters through multi-task fine-tuning. We do not perform fine-tuning separately for different tasks. For details of the training data, please refer to Appendix A.2.  The details of the training setup are included in Appendix A.4. Specifically, due to the significant variation in training data volume across different tasks, we conduct a three-stage training process. In the first training stage, the model is fine-tuned on all tasks using the complete training data as shown in Table 9. In the second stage, we continue fine-tuning the model on tasks that have small-scale datasets, including SER, SLU, AAC, TTS, and SE tasks.  In the third training stage, we continue fine-tuning the model again on all tasks using the complete training data in Table 9. We find that this design of training curriculum effectively reduces catastrophic forgetting (as demonstrated in competitive or superior performance on **all tasks**, please refer to our Global Response to Q1) and achieves high performance on tasks **with limited training data** (please refer to our Global Response to Q3).

---

> > ### Comment · Reviewer_Nqbz · 2023-12-04
> > **Response to rebuttal**
> >
> > I would like to thank the authors for their rebuttal. The clarification regarding AudioGPT is enlightening, thank you. I still hope that a slight elaboration like this can be added to the paper where it describes these systems. But still, I believe that the weaknesses outweigh the strengths and after reading the other reviews and the corresponding author comments, would still like to keep my rating and review as-is.

---

### Author Response · Authors · 2023-11-18
**Global Responses to All Reviewers (Part 1/4)**

We thank all the reviewers for your valuable comments and constructive suggestions, which are exceedingly helpful for us to improve our paper.

* We have uploaded a revised version of the paper to address some comments as illustrated in our responses.

* We posted detailed responses addressing all the weakness concerns and questions from each reviewer.

* Below are our responses to common questions shared by the reviewers.

***

**Q1. Performance comparisons of LauraGPT against baselines and SOTAs on each task**

**Response:**  We noticed that there are some misunderstandings and misinterpretations of the performance comparison between LauraGPT and the baselines on each task.  Below we summarize the performance comparisons for each task to demonstrate that **LauraGPT achieves competitive or superior performance compared to existing SOTA models on various audio processing benchmarks.**. First, we emphasize that **all baselines chosen for each task are mainstream models that are competitive and also widely evaluated on benchmarks used in our paper and on other datasets**, based on comprehensive literature review.

**For ASR task**,  the baselines are Paraformer and Whisper Large V2.  On the Chinese test sets,  LauraGPT greatly outperforms Whisper by **-3.9** and **-2.3** absolute on CER and performs comparably to Paraformer with a much smaller amount of training data. On the English test sets, LauraGPT achieves comparable performance to Paraformer and performs better on the more noisy test set, but performs worse than Whisper Large V2 as Whisper Large V2 uses much more English training data than LauraGPT. Note that there are other **targeted optimized models** [1] that can achieve better performance than Whisper Large V2 on LibriSpeech test sets. However, they only focus on English speech recognition and benefit from additional technologies such as language model decoding.

**For SLU task**, the baselines are CRDNN and Wav2Vec 2.0. LauraGPT significantly outperforms baselines on scenario/action/intent accuracies with **+1.55/+2.67/+2.53** absolute gains.  However, LauraGPT is slightly worse on slot filling (measured by Word-F1, Char-F1, and SLU-F1).  The possible explanations for the lower performance of LauraGPT on slot filling are twofold. (1) Slot filling requires fine-grained semantic parsing of the input utterance, whereas the current multi-task training tasks for LauraGPT do not emphasize fine-grained token-level semantic understanding.  Adding more tasks focusing on learning fine-grained semantic understanding into multi-task training, such as information extraction from speech and spoken question answering, may improve slot filling in SLU; (2)  Slot filling is a highly domain-specific task, which may not benefit sufficiently from the general knowledge of LauraGPT learned from multi-task learning. A possible solution is to explore domain-adaptive pre-training/fine-tuning of LauraGPT on slot filling.

**For S2TT task**,  we evaluate the performance on BSTC dev set (Zh→En) and CoVOST2 test set (En→Zh) and select the corresponding baselines for comparison. LauraGPT improves the baseline BLEU score by **+13.1** on the CoVOST2 test set (En→Zh) and achieves comparable performance on the BSTC dev set (Zh→En) with only  -0.4 BLEU reduction.

**For SER task**, we evaluate the MELD test set and select WavLM Base, WavLM Large and Vesper-12 for comparison. As described in [2], due to the sample imbalance of the MELD dataset, WF1 is the primary metric. LauraGPT outperforms all baselines on WF1. In terms of UA, LauraGPT also achieves the best performance.  Only in terms of WA, LauraGPT yields worse performance than WavLM Large and Vesper-12. The detailed analysis of these results is included in our Global Response to Q3, under the SER task.

**For AAC task**, as a single model, LauraGPT outperforms the baseline single-model EncDec-Attn on all metrics. Compared to the baseline Ensemble,  LauraGPT achieves a comparable SPICE score but underperforms on CIDEr and SPIDEr. SPICE is designed to capture the **specificity** and accuracy of the generated captions in terms of the semantic content, while CIDEr focuses on evaluating the **consensus** or agreement between the generated captions and the reference captions. SPIDEr can be viewed as an average of SPICE and CIDEr. Therefore, the results show that LauraGPT tends to generate captions that closely match one specific reference rather than the consensus of references.

---

> ### Author Response · Authors · 2023-11-18
> **Global Responses to All Reviewers (Part 2/4)**
>
> **For SE task**, LauraGPT slightly outperforms the **SOTA** CMGAN [3] in terms of **perceptual speech quality** (PESQ), while CMGAN  achieves better speech intelligibility in terms of STOI, CER and WER.  We attribute better speech intelligibility of CMGAN to its incorporation of multiple discriminators during training. We hypothesize that incorporating adversarial losses and fine-tuning the GPT backbone with the codec vocoder in end-to-end manner for LauraGPT would improve its speech intelligibility.
>
> **For TTS task**, baselines are the **SOTA** zero-shot speaker adaptive TTS model VALL-E. As the official implementation of VALL-E is not publicly available, we re-implement VALL-E models with 0.34B trainable parameters as competitive baselines. We train LauraGPT and baselines using the same training data for TTS task (Tabel 9) and evaluate on two zero-shot TTS test sets constructed using LibriTTS test-clean and AISHELL1-test corpora. Compared to baselines, LauraGPT achieves comparable speaker similarity and speech quality. Degradation on content consistency (CER/WER) from LauraGPT results from the generalization issue, since the training data is too limited for the large LauraGPT with 2B parameters.
>
> [1] C. S, et al. Wavlm: Large-scale self-supervised pre-training for full stack speech processing. IEEE Journal of Selected Topics in Signal Processing.
>
> [2] C. W, et al. Vesper: A compact and effective pretrained model for speech emotion recognition. arXiv:2307.10757.
>
> [3] R. Cao, et al. CMGAN: Conformer-based Metric GAN for Speech Enhancement. Interspeech 2022.
>
>
> ***
>
> **Q2: Performance comparisons between LauraGPT and the related multi-task unified audio-text models in Table 1**
>
> **Response:** Due to the drastic differences in experimental settings, datasets used and the lack of open source codebase and checkpoints, it is difficult to conduct a fair comparison between LauraGPT and the most related multi-task unified audio-text models in Table 1.  Despite all these difficulties, below we provide the most relevant results for comparing LauraGPT and these related models. Note that the Whisper model in Table 1 is solely studied on ASR in their paper and we have compared LauraGPT to Whisper, as shown in our Global Response to Q1.
>
> **SpeechT5** is evaluated on ASR, TTS, S2TT, voice conversion (VC), SE, and speaker identification (SID). Since the training data of tasks other than ASR for SpeechT5 differs remarkably from those for LauraGPT, we compare LauraGPT against SpeechT5 only on ASR.  For SpeechT5, the model is first pre-trained with large-scale unlabeled speech and text data. Then, it is finetuned on the Librispeech-960 corpus via the cross-entropy and CTC hybrid loss. As claimed in their paper, SpeechT5 achieves a WER of 7.3% on the test-other subset without CTC and LM. Under a fair comparison, our LauraGPT achieves a comparable WER of 7.7%.  Note that different from SpeechT5, LauraGPT is directly trained on multi-task labeled datasets without benefiting from any self-supervised pre-training.
>
> **VioLA** is evaluated on ASR, MT, S2TT, TTS and S2ST tasks. Considering the substantial differences in training data on tasks between VioLA and LauraGPT and the lack of open-sourced VioLA codebase/models, it is difficult to fairly compare LauraGPT with VioLA.  Among the tasks, direct comparison on ASR is also challenging since VioLA only conducts speech-to-phoneme recognition and reported Phoneme Error Rate (PER) rather than recognizing words/characters and reporting WER/CER as conducted by LauraGPT. According to their paper,  VioLA underperforms their in-house Attention-based Encoder-Decoder (AED) model (which we also have no access to) with relative 19.96% PER degradation from 9.47% to 11.36% on Mandarin WenetSpeech dev set. Since higher PER always corresponds to much higher WER as a word comprises multiple phonemes, it would be safe to hypothesize that the relative degradation on WER from VioLA over AED is even greater. In contrast, compared with the Paraformer baseline, our LauraGPT achieves comparable CER on the Mandarin AISHELL-2 test-ios set and outperforms it on the English Librispeech test-other set, i.e., overall LauraGPT performs comparably to Paraformer. Note that Paraformer is a non-autoregressive AED model performing comparably to conventional auto-regressive AED model [1]. Therefore, through this chain of comparisons, we consider LauraGPT notably outperforming VioLA on ASR task.
>
> **AudioPaLM** is evaluated on ASR, S2TT, TTS and MT tasks. Since the training and evaluation datasets for AudioPaLM and LauraGPT are disjoint, their performance results cannot be directly compared. In addition, the pre-trained model of AudioPaLM has not been released. Therefore, empirically comparing LauraGPT to AudioPaLM will require great effort and is not conducted in this work.
>
> [1] Zhifu Gao, et al. Paraformer: Fast and accurate parallel transformer for non-autoregressive end-to-end speech recognition. INTERSPEECH, 2022.

---

> ### Author Response · Authors · 2023-11-18
> **Global Responses to All Reviewers (Part 3/4)**
>
> ***
>
> **Q3. Value of the multi-task learning model.**
>
> **Response:**  The multi-task learned single model of LauraGPT has the following advantages over individual single-task models:
>
> 1. Multi-task learning could potentially exploit the synergy between related speech tasks and reduce over-fitting, hence LauraGPT could provide quality performance on a diverse set of tasks, and achieve better performance than single-task training, especially for tasks with limited training data.
>
> 2. Since multi-task learning could learn a single model capable of supporting a diverse set of tasks, it greatly simplifies the practical deployment and applications. LauraGPT can provide a diverse set of audio processing capabilities through the unified API and model implementation.
>
> The second point is straightforward.  For the first point, our Global Response to Q1 summarizes that **LauraGPT achieves competitive or superior performance compared to existing competitive and SOTA models on various audio processing benchmarks**. The following ablation analysis results further verify that **multi-task learning of LauraGPT achieves better performance than single-task training for tasks with limited training data**. For the AAC, SLU, and SER tasks, which all suffer from limited training data, we compare their single-task performance with the multi-task trained LauraGPT.   Specifically, we initialize the GPT with Qwen-LLM backbone and audio encoder with the same pre-trained checkpoints (i.e., the same initialization as LauraGPT before multi-task training), but train the model only using the AAC, SLU, and SER training data respectively.
>
> **For AAC task**, we find the multi-task trained LauraGPT outperforms the single-task model on all metrics, as shown in Table 16 in Appendix B.3.
>
> **For SLU task**, we find the multi-task trained LauraGPT outperforms the single-task model on scenario/action/intent accuracy with **+1.89/+3.11/+2.88** absolute gains, especially with **+21.22/+23.83/+22.46** absolute gains on slot filling Word-F1/Char-F1/SLU-F1, as shown in Table 17 in Appendix B.3.
>
> **For SER task**,  the multi-task trained LauraGPT significantly improves UA and WF1 over the single-task model, while the  WA result is slightly worse, as shown in Table 18 in Appendix B.3. Note that WF1 is the primary metric on MELD dataset due to the sample imbalance across different emotion classes [1]. To further analyze these results, we conduct a statistical analysis of the number of samples for each emotion class in both training and test sets of the MELD dataset, as well as their corresponding test accuracy, as shown in Table 19 in Appendix B.3.
>
> Our findings are as follows:
>
> 1. Compared to single-task training, multi-task learning results in a degradation in accuracy for classes with a larger number of training samples. However, multi-task learning significantly improves accuracy on classes with fewer training samples. This explains why WA decreases slightly from multi-task training, but UA and WF1 show significant improvements.
>
> 2. For single-task training,  the accuracy of the disgust and fear classes is 0, which aligns with the fact that these two classes have the fewest training samples in the MELD dataset.  Multi-task training not only remarkably improves the performance of emotion classes with low accuracy (joy, sadness, surprise), but also greatly improves the performance of classes that cannot be predicted with single-task training (disgust, fear).
>
> [1] Chen W, et al. Vesper: A compact and effective pretrained model for speech emotion recognition. arXiv:2307.10757.
>
> ***
>
> **Q4. Details of the training and inference processes**
>
> **Response:** As stated in Section 3.3, during the training stage, the input is converted into input embeddings by the audio encoder if the input is audio, or converted by the embedding matrix E if the input is text, while the output is converted into output embeddings by the same embedding matrix E for teacher-forcing. Meanwhile, this matrix E is also used to convert the task-ID token into an embedding. Then, these embeddings are composed into an embedding sequence as **[input embeddings, task-ID embedding, output embeddings]**, which is taken as the input of Qwen LLM. To train the model, a masked cross-entropy loss function is applied, as shown in Eq. (1).  As described in Section 3.5,  in addition to masking out the losses on inputs, the cross-entropy loss at the position of the task token is also masked out.
>
> During the inference stage, the input is converted into input embeddings as done during the training stage. Then the corresponding task-ID embedding is added at the end of the input embedding sequence. Next, the Qwen LLM generates output tokens in an autoregressive manner until the <eos> token is generated. Finally, for text-format output,  the Qwen tokenizer is employed to convert tokens into final output, while for audio-format output, the codec vocoder is employed to convert tokens into raw waveforms.

---

> ### Author Response · Authors · 2023-11-18
> **Global Responses to All Reviewers (Part 4/4)**
>
> ***
>
> **Q5: Two different taxonomies of related models are introduced in Section 1 and 2**
>
> **Response:**  As described in Section 2, existing works on unified audio-text modeling can be categorized into four groups: (1) self-supervised learning of a universal audio encoder, (2) encoder-decoder models with modal-specific pre-nets and post-nets, (3) models converting audio features to text (including encoder-decoder and decoder-only models), and (4) decoder-only models with discrete audio tokens. In Section 1, we only mentioned group (3) models converting audio features to text (including encoder-decoder and decoder-only models) and group (4) decoder-only models with discrete audio tokens, since these two categories of prior works are **most related to our work**.  Then in Section 2, we dive into more details of all four categories. We have updated the paper to clarify this point and make it clear that we use one taxonomy of related models consistently in the paper.
>
> ***
>
> **Q6: A few figures going into more detail for each of the components in Figure 1 would also greatly improve the readability of the the method section. Currently, Figure 1 is very high-level and does not offer the reader too much.**
>
> **Response:** We have updated Figure 1 in the revised version to demonstrate the unified task expression, the model architecture, and details of the codec vocoder.

---

### Author Response · Authors · 2023-11-21
**Gentle Reminder to Please Read Our Author Responses**

We thank all of the reviewers for your valuable comments and constructive suggestions.

* We have responded to the common questions shared by reviewers in our **Global Responses to All Reviewers**.
* We have responded to **each reviewer** separately by addressing any remaining concerns and questions not covered by Global Responses to All Reviewers.
* We have **strengthened our submission** with additional experimental results, discussions and analyses, updated figure, clarification and details based on the reviewer comments, and **updated the paper**. We have clarified the revisions in our responses.

Combining **Global Responses to All Reviewers** and respective responses to each reviewer, we believe that we have addressed all reviewer concerns and questions on November 18.

Would reviewers please read our author responses? Please let us know if there are any other questions or concerns during the discussion period. Thank you!

---

### Comment · Area_Chair_i2q2 · 2023-11-21
**Reminder to reviewers to participate in the author/reviewer discussion**

Dear reviewers, this is a reminder that the author/reviewer discussion period ends November 22.

This discussion is indeed supposed to be a dialog, so please respond to
the comments from the authors as well as the updated manuscript.

AC

---

### Meta-Review · Area_Chair_i2q2 · 2023-12-07

**Metareview:**

## Scientific Claims and Findings
This paper describes a generative pre-trained transformer (decoder-only) model that can perform a wide range of audio and natural language processing tasks. Like some other recent models in this area such as SpeechGPT, VioLA, and AudioPaLM, it is capable of processing input from both text and audio sources and can generate both text and audio outputs. Aspects of the proposed model that distinguish it from prior work include (1) the use of continuous audio representations as input to the LM and (2) one-step generation of audio using only the coarsest quantizer's output as audio tokens plus audio or text conditioning information. The paper benchmarks the model on a range of speech and text tasks: automatic speech recognition, spoken language understanding, speech-to-ttext translation, speech emotion recognition, audio captioning, speech enhancement, text-to-speech, and (in the supplemental material) machine translation. The evaluations show that the model performs on par with competing models, many of which are single-task models. A comparison of using discrete audio tokens as input instead of continuous vectors shows that the continuous representations lead to better performance.

## Strengths
- The breadth of tasks that the model is capable of and the breadth of evaluation are impressive.

## Weaknesses
- The work is somewhat incremental, in the sense that it has already been clearly demonstrated that this style of model can perform a wide range of audio and text tasks. The main contribution of the paper is doing additional tasks.

- [From the AC] I have serious concerns about the experiments designed to show that using continuous audio representations as inputs is necessary to achieve good performance. The gaps reported in this paper are much larger than those reported in a recent paper, which I am aware is contemporaneous work, https://arxiv.org/abs/2309.10922, and aren't consistent with the results reported with models such as AudioPaLM, which appear to work reasonably well using discrete representations of audio. The description of the discrete/continuous contrast experiment is quite shallow, which makes it more difficult to interpret these results. For example, it is not clear what training data was used to train the audio tokenizer. Also, looking at the tables of results showing the number of trainable parameters, it is apparent that there are 200M parameters in the audio encoder, and according to Section 4, the audio encoder is fine tuned on the various tasks. A very helpful contrast experiment would use continuous audio representations, but would not fine-tune the encoder.

- The state of the art on the SLURP task was achieved using fine-tuned HuBERT models in Y. Wang, A. Boumadane, and A. Heba, “A fine-tuned wav2vec 2.0/HuBERT benchmark for speech emotion recognition, speaker verification and spoken language understanding,” arXiv preprint arXiv:2111.02735, 2021. They are better than 89% intent classification accuracy and better than 78% slot-filling F1 score, and the LauraGPT results lag this by a lot.

- The description of the SLU task in the paper is inconsistent: it is described in A.1 as the task of "deriving high-level semantic meaning directly from audio input", but in Section 3.5 it is stated that the "SLU task take[s] test embeddings as inputs."

- Discarding all quantizer outputs except the first in the audio tokenizer is an unusual choice that I would expect would lead to poor quality in the generated audio, but the MOSNet results in Table 8 seem to indicate this is not an issue. The only way I can explain this result is through the provision of the text and audio conditioning to the audio decoder. The paper would be a lot clearer if this issue were discussed in more detail and if there were an ablation of the conditioning information to demonstrate its importance.

## Questions
- How large of an input context is supported by the Qwen model? I ask because it is unusual in this style of model to use a rate of 100 per second for audio representations, as this will rapidly fill the input context.

**Justification For Why Not Higher Score:**

I am inclined to reject the paper, primarily because I don't believe the experimental results contrasting the continuous and discrete audio representations, as I discuss above under "Weaknesses", but also because there are some issues with the clarity of the paper in the description of how the tokenizer was trained and the importance of the conditioning information in achieving good audio generation with only coarse tokens.

**Justification For Why Not Lower Score:**

N/A

---

### Decision · Program_Chairs · 2024-01-16

Reject